# ON SOLVING MINIMAX OPTIMIZATION LOCALLY: A FOLLOW-THE-RIDGE APPROACH

**Yuanhao Wang**[*1], **Guodong Zhang**[*2,3], **Jimmy Ba**[2,3]
[1]IIIS, Tsinghua University, [2]University of Toronto, [3]Vector Institute
`yuanhao-16@mails.tsinghua.edu.cn, {gdzhang,jba}@cs.toronto.edu`

## ABSTRACT

Many tasks in modern machine learning can be formulated as finding equilibria in *sequential* games. In particular, two-player zero-sum sequential games, also known as minimax optimization, have received growing interest. It is tempting to apply gradient descent to solve minimax optimization given its popularity and success in supervised learning. However, it has been noted that naive application of gradient descent fails to find some local minimax and can converge to non-local-minimax points. In this paper, we propose *Follow-the-Ridge* (FR), a novel algorithm that provably converges to and only converges to local minimax. We show theoretically that the algorithm addresses the notorious rotational behaviour of gradient dynamics, and is compatible with preconditioning and *positive* momentum. Empirically, FR solves toy minimax problems and improves the convergence of GAN training compared to the recent minimax optimization algorithms[1].

## 1 INTRODUCTION

We consider differentiable *sequential* games with two players: a leader who can commit to an action, and a follower who responds after observing the leader's action. Particularly, we focus on the zero-sum case of this problem which is also known as minimax optimization, *i.e.*,

$$\min_{\mathbf{x} \in \mathbb{R}^n} \max_{\mathbf{y} \in \mathbb{R}^m} f(\mathbf{x}, \mathbf{y}).$$

Unlike simultaneous games, many practical machine learning algorithms, including generative adversarial networks (GANs) (Goodfellow et al., 2014; Arjovsky et al., 2017), adversarial training (Madry et al., 2018) and primal-dual reinforcement learning (Du et al., 2017; Dai et al., 2018), explicitly specify the order of moves between players and the order of which player acts first is crucial for the problem. Therefore, the classical notion of local Nash equilibrium from simultaneous games may not be a proper definition of local optima for sequential games since mini-

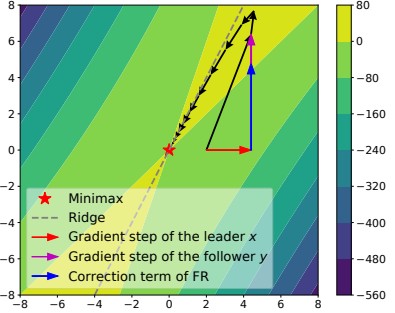

**Figure 1:** For a quadratic function $f(x, y) = -3x^2 + 4xy - y^2$, our algorithm moves closer to the ridge every iteration and it moves along the ridge once it hits the ridge. Without the FR correction term, gradient dynamics can drift away from the ridge.

max is in general not equal to maximin. Instead, we consider the notion of *local minimax* (Jin et al., 2019) which takes into account the sequential structure of minimax optimization.

The vanilla algorithm for solving sequential minimax optimization is gradient descent-ascent (GDA), where both players take a gradient update simultaneously. However, GDA is known to suffer from two drawbacks. First, it has undesirable convergence properties: it fails to converge to some local minimax and can converge to fixed points that are not local minimax (Jin et al., 2019; Daskalakis and Panageas, 2018). Second, GDA exhibits strong rotation around fixed points, which requires using very small learning rates (Mescheder et al., 2017; Balduzzi et al., 2018) to converge.

In this paper, we propose *Follow-the-Ridge* (FR), an algorithm for minimax optimization that addresses both issues. Specifically, we elucidate the cause of undesirable convergence of GDA – the

---

[*]These two authors contributed equally.
[1]Our code is made public at: `https://github.com/gd-zhang/Follow-the-Ridge`

leader whose gradient step takes the system away from the ridge. By adding a correction term to the follower, we explicitly cancel out negative effects of the leader's update. Intuitively, the combination of the leader's update and the correction term is parallel to the ridge in the landscape (see Fig. 1), hence the name *Follow-the-Ridge*. Overall, our contributions are the following:

- We propose a novel algorithm for minimax optimization which has exact local convergence to local minimax points. Previously, this property was only known to be satisfied when the leader moves infinitely slower than the follower in gradient descent-ascent (Jin et al., 2019).

- We show theoretically and empirically that FR addresses the notorious rotational behaviour of gradient dynamics around fixed points (Balduzzi et al., 2018) and thus allows a much larger learning rate compared to GDA.

- We prove that our algorithm is compatible with standard acceleration techniques such as preconditioning and *positive* momentum, which can speed up convergence significantly.

- We further show that our algorithm also applies to general-sum Stackelberg games (Fiez et al., 2019; Zeuthen, 1935) with similar theoretical guarantees.

- Finally, we demonstrate empirically our algorithm improves the convergence performance in both toy minimax problems and GAN training compared to existing methods.

## 2 PRELIMINARIES

### 2.1 MINIMAX OPTIMIZATION

We consider sequential games with two players where one player is deemed the *leader* and the other the *follower*. We denote leader's action by $\mathbf{x} \in \mathbb{R}^n$, and the follower's action by $\mathbf{y} \in \mathbb{R}^m$. The leader aims at minimizing the cost function $f(\mathbf{x}, \mathbf{y})$ while the follower aims at maximizing $f(\mathbf{x}, \mathbf{y})$. The only assumption we make on the cost function is the following.

**Assumption 1.** *$f$ is twice differentiable everywhere, and thrice differentiable at critical points. $\nabla_{\mathbf{yy}}^2 f$ is invertible (i.e., non-singular).*

The global solution to the sequential game $\min_{\mathbf{x}} \max_{\mathbf{y}} f(\mathbf{x}, \mathbf{y})$ is an action pair $(\mathbf{x}^*, \mathbf{y}^*)$, such that $\mathbf{y}^*$ is the global optimal response to $\mathbf{x}^*$ for the follower, and that $\mathbf{x}^*$ is the global optimal action for the leader assuming the follower always play the global optimal response. We call this global solution the *global minimax*. However, finding this global minimax is often intractable; therefore, we follow Jin et al. (2019) and take *local minimax* as the local surrogate.

**Definition 1** (local minimax). *$(\mathbf{x}^*, \mathbf{y}^*)$ is a local minimax for $f(\mathbf{x}, \mathbf{y})$ if (1) $\mathbf{y}^*$ is a local maximum of $f(\mathbf{x}^*, \cdot)$; (2) $\mathbf{x}^*$ is a local minimum of $\phi(\mathbf{x}) := f(\mathbf{x}, r(\mathbf{x}))$, where $r(\mathbf{x})$ is the implicit function defined by $\nabla_{\mathbf{y}} f(\mathbf{x}, \mathbf{y}) = 0$ in a neighborhood of $\mathbf{x}^*$ with $r(\mathbf{x}^*) = \mathbf{y}^*$.*

In the definition above, the implicit function $r(\cdot) : \mathbb{R}^n \to \mathbb{R}^m$ is a local best response for the follower, and is a *ridge* in the landscape of $f(\mathbf{x}, \mathbf{y})$. Local minimaxity captures an equilibrium in a two-player sequential game if both players are only allowed to change their strategies locally. For notational convenience, we define

$$\nabla f(\mathbf{x}, \mathbf{y}) = [\nabla_{\mathbf{x}} f, \nabla_{\mathbf{y}} f]^\top, \ \nabla^2 f(\mathbf{x}, \mathbf{y}) = \begin{bmatrix} \mathbf{H_{xx}} & \mathbf{H_{xy}} \\ \mathbf{H_{yx}} & \mathbf{H_{yy}} \end{bmatrix}.$$

In principle, local minimax can be characterized in terms of the following first-order and second-order conditions, which were established in Jin et al. (2019).

**Proposition 1** (First-order Condition). *Any local minimax $(\mathbf{x}^*, \mathbf{y}^*)$ satisfies $\nabla f(\mathbf{x}^*, \mathbf{y}^*) = 0$.*

**Proposition 2** (Second-order Necessary Condition). *Any local minimax $(\mathbf{x}^*, \mathbf{y}^*)$ satisfies $\mathbf{H_{yy}} \preccurlyeq 0$ and $\mathbf{H_{xx}} - \mathbf{H_{xy}} \mathbf{H_{yy}}^{-1} \mathbf{H_{yx}} \succcurlyeq 0$.*

**Proposition 3** (Second-order Sufficient Condition). *Any stationary point $(\mathbf{x}^*, \mathbf{y}^*)$ satisfying $\mathbf{H_{yy}} \prec 0$ and $\mathbf{H_{xx}} - \mathbf{H_{xy}} \mathbf{H_{yy}}^{-1} \mathbf{H_{yx}} \succ 0$ is a local minimax.*

The concept of global/local minimax is different from *Nash equilibrium* and *local Nash*, which are the equilibrium concepts typically studied for simultaneous games (see Nash et al. (1950); Ratliff et al. (2016) for more details). In particular, we note that the concept of Nash equilibrium or local Nash does not reflect the order between the min-player and the max-player and may not exist even

for simple functions (Jin et al., 2019). In general, the set of local minimax is a superset of local Nash. Under some mild assumptions, local minimax points are guaranteed to exist (Jin et al., 2019). However, the set of stable fixed points of GDA, roughly speaking the set of points that GDA locally converges to, is a different superset of local Nash (Jin et al., 2019). The relation between the three sets of points is illustrated in Fig. 2.

## 2.2 STABILITY OF DISCRETE DYNAMICAL SYSTEMS

Gradient-based methods can reliably find local stable fixed points – local minima in single-objective optimization. Here, we generalize the concept of stability to games by taking game dynamics as a discrete dynamical system. An iteration of the form $\mathbf{z}_{t+1} = w(\mathbf{z}_t)$ can be viewed as a discrete dynamical system, where in our case $w : \mathbb{R}^{n+m} \to \mathbb{R}^{n+m}$. If $w(\mathbf{z}) = \mathbf{z}$, then $\mathbf{z}$ is called a fixed point. We study the stability of fixed points as a proxy to local convergence of game dynamics.

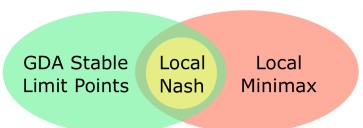

**Figure 2:** Relation between local Nash, local minimax and GDA stable fixed points.

**Definition 2.** *Let $\mathbf{J}$ denote the Jacobian of $w$ at a fixed point $\mathbf{z}$. If it has spectral radius $\rho(\mathbf{J}) \leq 1$, then we call $\mathbf{z}$ a stable fixed point. If $\rho(\mathbf{J}) < 1$, then we call $\mathbf{z}$ a strictly stable fixed point.*

It is known that strict stability implies local convergence (*e.g.*, see Galor (2007)). In other words, if $\mathbf{z}$ is a strictly stable fixed point, there exists a neighborhood $U$ of $\mathbf{z}$ such that when initialized in $U$, the iteration steps always converge to $\mathbf{z}$.

## 3 UNDESIRABLE BEHAVIOURS OF GDA

In this section, we discuss the undesirable behaviours of GDA in more detail. Recall that the update rule of GDA is given by

$$\begin{aligned} \mathbf{x}_{t+1} &\leftarrow \mathbf{x}_t - \eta \nabla_{\mathbf{x}} f, \\ \mathbf{y}_{t+1} &\leftarrow \mathbf{y}_t + \eta \nabla_{\mathbf{y}} f, \end{aligned} \tag{1}$$

where we assume the same learning rate for both the leader and the follower for simplicity[2]. As illustrated in Fig. 2, the set of stable fixed points of GDA can include points that are not local minimax and, perhaps even worse, some local minimax are not necessarily stable fixed points of GDA. Here, we first give an example that a stable fixed point of GDA is not a local minimax. Consider $\min_x \max_y f(x, y) = 3x^2 + y^2 + 4xy$; the only stationary point of this problem is $(0, 0)$ and the Jacobian of GDA at this point is

$$\mathbf{J} = \mathbf{I} - \eta \begin{bmatrix} 6 & 4 \\ -4 & -2 \end{bmatrix}.$$

It is easy to see that the eigenvalues of $\mathbf{J}$ are $e_1 = e_2 = 1 - 2\eta$. Therefore, by Definition 2, $(0, 0)$ is a strictly stable fixed point of GDA. However, one can show that $\mathbf{H_{yy}} = 2 > 0$ which doesn't satisfy the second-order necessary condition of local minimax.

Similarly, one can easily find examples in which a local minimax is not in the set of stable fixed points of GDA, *e.g.*, $\min_{x \in \mathbb{R}} \max_{y \in \mathbb{R}} f(x, y) = -3x^2 - y^2 + 4xy$ (see Fig. 1). In this example, the two Jacobian eigenvalues are both greater than 1 no matter how small the learning rate is. In other words, GDA fails to converge to $(0, 0)$ for almost all initializations (Daskalakis and Panageas, 2018).

As we will discuss in the next section, the main culprit of the undesirable behaviours of GDA is the leader whose gradient update $-\eta \nabla_{\mathbf{x}} f$ pushes the whole system away from the ridge or attracts the system to non-local-minimax points. By contrast, the follower's step $\eta \nabla_{\mathbf{y}} f$ can pull the system closer to the ridge (see Fig. 1) or push it away from bad fixed points. To guarantee convergence to local minimax (or avoid bad fixed points), we have to use a very small learning rate for the leader (Jin et al., 2019; Fiez et al., 2019) so that the $\eta \nabla_{\mathbf{y}} f$ term dominates. In the next section, we offer an alternative approach which explicitly cancels out undesirable effects of $-\eta \nabla_{\mathbf{x}} f$, thereby allowing us to use larger learning rates for the leader.

---

[2]In general, the learning rates of two players can be different. Since our arguments apply to general setting as long as the ratio $\eta_{\mathbf{x}} / \eta_{\mathbf{y}}$ is a positive constant, so we assume the same learning rate for convenience.

## 4    FOLLOW THE RIDGE

Despite its popularity, GDA has the tendency to drift away from the ridge or the implicit function, and can, therefore, fail to converge with any constant learning rate. To address these problems, we propose a novel algorithm for minimax optimization, which we term *Follow-the-Ridge* (FR). The algorithm modifies gradient descent-ascent by applying an asymmetric preconditioner. The update rule is described in Algorithm. 1.

---

**Algorithm 1** Follow-the-Ridge (FR). Differences from gradient descent-ascent are shown in blue.

---

**Require:** Learning rate $\eta_{\mathbf{x}}$ and $\eta_{\mathbf{y}}$; number of iterations $T$.
1: **for** $t = 1, ..., T$ **do**
2:     $\mathbf{x}_{t+1} \leftarrow \mathbf{x}_t - \eta_{\mathbf{x}} \nabla_{\mathbf{x}} f(\mathbf{x}_t, \mathbf{y}_t)$                                                                ▷ gradient descent
3:     $\mathbf{y}_{t+1} \leftarrow \mathbf{y}_t + \eta_{\mathbf{y}} \nabla_{\mathbf{y}} f(\mathbf{x}_t, \mathbf{y}_t) + \eta_{\mathbf{x}} \mathbf{H}_{\mathbf{yy}}^{-1} \mathbf{H}_{\mathbf{yx}} \nabla_{\mathbf{x}} f(\mathbf{x}_t, \mathbf{y}_t)$          ▷ modified gradient ascent

---

The main intuition behind FR is the following. Suppose that $\mathbf{y}_t$ is a local maximum of $f(\mathbf{x}_t, \cdot)$. Let $r(\mathbf{x})$ be the implicit function defined by $\nabla_{\mathbf{y}} f(\mathbf{x}, \mathbf{y}) = 0$ around $(\mathbf{x}_t, \mathbf{y}_t)$, *i.e.*, a ridge in the landscape of $f(\mathbf{x}, \mathbf{y})$. By definition, a local minimax has to lie on a ridge; hence, it is intuitive to follow the ridge during learning. However, if $(\mathbf{x}_t, \mathbf{y}_t)$ is on the ridge, then $\nabla_{\mathbf{y}} f(\mathbf{x}_t, \mathbf{y}_t) = 0$, and one step of gradient descent-ascent will take $(\mathbf{x}_t, \mathbf{y}_t)$ to $(\mathbf{x}_t - \eta_{\mathbf{x}} \nabla_{\mathbf{x}} f, \mathbf{y}_t)$, which is off the ridge. In other words, gradient descent-ascent tends to drift away from the ridge. The correction term we introduce is

$$\nabla_{\mathbf{x}} r(\mathbf{x}) \left( -\eta_{\mathbf{x}} \nabla_{\mathbf{x}} f(\mathbf{x}_t, \mathbf{y}_t) \right) = \eta_{\mathbf{x}} \mathbf{H}_{\mathbf{yy}}^{-1} \mathbf{H}_{\mathbf{yx}} \nabla_{\mathbf{x}} f.$$

It would bring $\mathbf{y}_t$ to $\mathbf{y}_t + \nabla_{\mathbf{x}} r(\mathbf{x})(\mathbf{x}_{t+1} - \mathbf{x}_t) \approx r(\mathbf{x}_{t+1})$, thereby encouraging both players to stay along the ridge. When $(\mathbf{x}_t, \mathbf{y}_t)$ is not on a ridge yet, we expect the $-\eta_{\mathbf{x}} \nabla_{\mathbf{x}} f$ term and the $\eta_{\mathbf{x}} \mathbf{H}_{\mathbf{yy}}^{-1} \mathbf{H}_{\mathbf{yx}} \nabla_{\mathbf{x}} f$ term to move parallel to the ridge, while the $\eta_{\mathbf{y}} \nabla_{\mathbf{y}} f$ term brings $(\mathbf{x}_t, \mathbf{y}_t)$ closer to the ridge (see Fig. 1). Our main theoretical result is the following theorem, which suggests that FR locally converges and only converges to local minimax.

**Theorem 1** (Exact local convergence). *With a suitable learning rate, all strictly stable fixed points of FR are local minimax, and all local minimax points are stable fixed points of FR.*

The proof is mainly based on the following observation. The Jacobian of FR dynamics at a fixed point $(\mathbf{x}^*, \mathbf{y}^*)$ is ($c := \eta_{\mathbf{y}}/\eta_{\mathbf{x}}$)

$$\mathbf{J} = \mathbf{I} - \eta_{\mathbf{x}} \begin{bmatrix} \mathbf{I} \\ -\mathbf{H}_{\mathbf{yy}}^{-1}\mathbf{H}_{\mathbf{yx}} & \mathbf{I} \end{bmatrix} \begin{bmatrix} \mathbf{H}_{\mathbf{xx}} & \mathbf{H}_{\mathbf{xy}} \\ -c\mathbf{H}_{\mathbf{yx}} & -c\mathbf{H}_{\mathbf{yy}} \end{bmatrix},$$

where the Hessians are evaluated at $(\mathbf{x}^*, \mathbf{y}^*)$. $\mathbf{J}$ is similar to

$$\mathbf{M} = \begin{bmatrix} \mathbf{I} \\ \mathbf{H}_{\mathbf{yy}}^{-1}\mathbf{H}_{\mathbf{yx}} & \mathbf{I} \end{bmatrix} \mathbf{J} \begin{bmatrix} \mathbf{I} \\ -\mathbf{H}_{\mathbf{yy}}^{-1}\mathbf{H}_{\mathbf{yx}} & \mathbf{I} \end{bmatrix} = \mathbf{I} - \eta_{\mathbf{x}} \begin{bmatrix} \mathbf{H}_{\mathbf{xx}} - \mathbf{H}_{\mathbf{xy}}\mathbf{H}_{\mathbf{yy}}^{-1}\mathbf{H}_{\mathbf{yx}} & \mathbf{H}_{\mathbf{xy}} \\ & -c\mathbf{H}_{\mathbf{yy}} \end{bmatrix}.$$

Therefore, the eigenvalues of $\mathbf{J}$ are those of $\mathbf{I} + \eta_{\mathbf{y}} \mathbf{H}_{\mathbf{yy}}$ and those of $\mathbf{I} - \eta_{\mathbf{x}} (\mathbf{H}_{\mathbf{xx}} - \mathbf{H}_{\mathbf{xy}}\mathbf{H}_{\mathbf{yy}}^{-1}\mathbf{H}_{\mathbf{yx}})$. As shown in second-order necessary condition 2, $(\mathbf{x}^*, \mathbf{y}^*)$ being a local minimax implies $\mathbf{H}_{\mathbf{yy}} \preccurlyeq 0$ and $\mathbf{H}_{\mathbf{xx}} - \mathbf{H}_{\mathbf{xy}}\mathbf{H}_{\mathbf{yy}}^{-1}\mathbf{H}_{\mathbf{yx}} \succcurlyeq 0$; one can then show that the spectral radius of the Jacobian satisfies $\rho(\mathbf{J}) \leq 1$; hence $(\mathbf{x}^*, \mathbf{y}^*)$ is a stable fixed point by Definition 2. On the other hand, when $\rho(\mathbf{J}) < 1$, by the sufficient condition in Proposition 3, $(\mathbf{x}^*, \mathbf{y}^*)$ must be a local minimax.

**Remark 1** (All eigenvalues are real). *We notice that all eigenvalues of $\mathbf{J}$, the Jacobian of FR, are real since both $\mathbf{H}_{\mathbf{yy}}$ and $\mathbf{H}_{\mathbf{xx}} - \mathbf{H}_{\mathbf{xy}}\mathbf{H}_{\mathbf{yy}}^{-1}\mathbf{H}_{\mathbf{yx}}$ are symmetric matrices. As noted by Mescheder et al. (2017); Gidel et al. (2019); Balduzzi et al. (2018), the rotational behaviour (instability) of GDA is caused by eigenvalues with large imaginary part. Therefore, FR addresses the strong rotation problem around fixed points as all eigenvalues are real.*

### 4.1    ACCELERATING CONVERGENCE WITH PRECONDITIONING AND MOMENTUM

We now discuss several extension of FR that preserves the theoretical guarantees.

**Preconditioning:** To speed up the convergence, it is often desirable to apply a preconditioner on the gradients that compensates for the curvature. For FR, the preconditioned variant is given by

$$\begin{bmatrix} \mathbf{x}_{t+1} \\ \mathbf{y}_{t+1} \end{bmatrix} \leftarrow \begin{bmatrix} \mathbf{x}_t \\ \mathbf{y}_t \end{bmatrix} - \begin{bmatrix} \mathbf{I} \\ -\mathbf{H}_{\mathbf{yy}}^{-1}\mathbf{H}_{\mathbf{yx}} & \mathbf{I} \end{bmatrix} \begin{bmatrix} \eta_{\mathbf{x}} \mathbf{P}_1 \nabla_{\mathbf{x}} f \\ -\eta_{\mathbf{y}} \mathbf{P}_2 \nabla_{\mathbf{y}} f \end{bmatrix} \tag{2}$$

We can show that with *any* constant positive definite preconditioners $\mathbf{P}_1$ and $\mathbf{P}_2$, the local convergence behavior of Algorithm 1 remains exact. We note that preconditioning is crucial for successfully training GANs (see Fig. 9) and RMSprop/Adam has been exclusively used in GAN training.

**Momentum:** Another important technique in optimization is momentum, which speeds up convergence significantly both in theory and in practice (Polyak, 1964; Sutskever et al., 2013). We show that momentum can be incorporated into FR (here, we include momentum outside the correction term which is equivalent to applying momentum to the gradient directly for simplicity. We give a detailed discussion in Appendix D.4), which gives the following update rule:

$$\begin{bmatrix} \mathbf{x}_{t+1} \\ \mathbf{y}_{t+1} \end{bmatrix} \leftarrow \begin{bmatrix} \mathbf{x}_t \\ \mathbf{y}_t \end{bmatrix} - \begin{bmatrix} \mathbf{I} & \\ -\mathbf{H}_{\mathbf{yy}}^{-1}\mathbf{H}_{\mathbf{yx}} & \mathbf{I} \end{bmatrix} \begin{bmatrix} \eta_{\mathbf{x}}\nabla_{\mathbf{x}}f \\ -\eta_{\mathbf{y}}\nabla_{\mathbf{y}}f \end{bmatrix} + \gamma \begin{bmatrix} \mathbf{x}_t - \mathbf{x}_{t-1} \\ \mathbf{y}_t - \mathbf{y}_{t-1} \end{bmatrix}. \tag{3}$$

Because all of the Jacobian eigenvalues are real, we can show that momentum speeds up local convergence in a similar way it speeds up single objective minimization.

**Theorem 2.** *For local minimax* $(\mathbf{x}^*, \mathbf{y}^*)$*, let* $\alpha = \min\left\{\lambda_{min}(-\mathbf{H}_{\mathbf{yy}}), \lambda_{min}(\mathbf{H}_{\mathbf{xx}} - \mathbf{H}_{\mathbf{xy}}\mathbf{H}_{\mathbf{yy}}^{-1}\mathbf{H}_{\mathbf{yx}})\right\}$, $\beta = \rho\left(\nabla^2 f(\mathbf{x}^*, \mathbf{y}^*)\right)$, $\kappa := \beta/\alpha$. Then FR converges asymptotically to $(\mathbf{x}^*, \mathbf{y}^*)$ with a rate $\Omega(\kappa^{-2})$; FR with a momentum parameter of $\gamma = 1 - \Theta\left(\kappa^{-1}\right)$ converges asymptotically with a rate $\Omega(\kappa^{-1})$.*[3]

Experiments of the speedup of momentum are provided in Appendix E.2. This is in contrast to gradient descent-ascent, whose complex Jacobian eigenvalues prevent the use of positive momentum. Instead, negative momentum may be more preferable (Gidel et al., 2019), which does not achieve the same level of acceleration.

## 4.2 GENERAL STACKELBERG GAMES

---

**Algorithm 2** Follow-the-Ridge (FR) for general-sum Stackelberg games.

---

**Require:** Learning rate $\eta_{\mathbf{x}}$ and $\eta_{\mathbf{y}}$; number of iterations $T$.
1: **for** $t = 1, ..., T$ **do**
2:     $\mathbf{x}_{t+1} \leftarrow \mathbf{x}_t - \eta_{\mathbf{x}}D_{\mathbf{x}}f(\mathbf{x}_t, \mathbf{y}_t)$         ▷ total derivative $D_{\mathbf{x}}f = \nabla_{\mathbf{x}}f - \nabla^2_{\mathbf{xy}}g(\nabla^2_{\mathbf{yy}}g)^{-1}\nabla_{\mathbf{y}}f$
3:     $\mathbf{y}_{t+1} \leftarrow \mathbf{y}_t - \eta_{\mathbf{y}}\nabla_{\mathbf{y}}g(\mathbf{x}_t, \mathbf{y}_t) + \eta_{\mathbf{x}}(\nabla^2_{\mathbf{yy}}g)^{-1}\nabla^2_{\mathbf{yx}}gD_{\mathbf{x}}f(\mathbf{x}_t, \mathbf{y}_t)$

---

Here, we further extend FR to general sequential games, also known as Stackelberg games. The leader commits to an action $\mathbf{x}$, while the follower plays $\mathbf{y}$ in response. The leader aims to minimize its cost $f(\mathbf{x}, \mathbf{y})$, while the follower aims at minimizing $g(\mathbf{x}, \mathbf{y})$. For Stackelberg games, the notion of equilibrium is captured by *Stackelberg equilibrium*, which is essentially the solution to the following optimization problem:

$$\min_{\mathbf{x}\in\mathbb{R}^n}\left\{f(\mathbf{x}, \mathbf{y}) | \mathbf{y} \in \arg\min_{\mathbf{y}\in\mathbb{R}^m} g(\mathbf{x}, \mathbf{y})\right\}.$$

It can be seen that minimax optimization is the special case when $g = -f$.

Similarly, one can define local Stackelberg equilibrium as a generalization of local minimax in general-sum games (Fiez et al., 2019). Stackelberg game has wide applications in machine learning. To name a few, both multi-agent reinforcement learning (Littman, 1994) and hyperparameter optimization (Maclaurin et al., 2015) can be formulated as finding Stackelberg equilibria.

For general-sum games, naive gradient dynamics, *i.e.*, both players taking gradient updates with their own cost functions, is no longer a reasonable algorithm, as local Stackelberg equilibria in general may not be stationary points. Instead, the leader should try to use the total derivative of $f(\mathbf{x}, r(\mathbf{x}))$, where $r(\mathbf{x})$ is a local best response for the follower. Thus the counterpart of gradient descent-ascent in general-sum games is actually gradient dynamics with best-response gradient (Fiez et al., 2019):

$$\begin{aligned} \mathbf{x}_{t+1} &\leftarrow \mathbf{x}_t - \eta\left[\nabla_{\mathbf{x}}f - \nabla^2_{\mathbf{xy}}g\left(\nabla^2_{\mathbf{yy}}g\right)^{-1}\nabla_{\mathbf{y}}f\right](\mathbf{x}_t, \mathbf{y}_t), \\ \mathbf{y}_{t+1} &\leftarrow \mathbf{y}_t - \eta\nabla_{\mathbf{y}}g(\mathbf{x}_t, \mathbf{y}_t). \end{aligned} \tag{4}$$

---

[3]By a rate $a$, we mean that one iteration shortens the distance toward the fixed point by a factor of $(1 - a)$; hence the larger the better.

FR can be adapted to general-sum games by adding the same correction term to the follower. The combined update rule is given in Algorithm 2. Similarly, we show that FR for Stackelberg games locally converges exactly to local Stackelberg equilibria (see Appendix C.2 for rigorous proof.)

## 5 RELATED WORK

As a special case of Stackelberg games (Ratliff et al., 2016) in the zero-sum setting, minimax optimization concerns the problem of solving $\min_{\mathbf{x} \in \mathcal{X}} \max_{\mathbf{y} \in \mathcal{Y}} f(\mathbf{x}, \mathbf{y})$. The problem has received wide attention due to its extensive applications in modern machine learning, in settings such as generative adversarial networks (GANs), adversarial training. The vast majority of this line of research focus on convex-concave setting (Kinderlehrer and Stampacchia, 1980; Nemirovski and Yudin, 1978; Nemirovski, 2004; Mokhtari et al., 2019b;a). Beyond the convex-concave setting, Rafique et al. (2018); Lu et al. (2019); Lin et al. (2019); Nouiehed et al. (2019) consider nonconvex-concave problems, *i.e.*, where $f$ is nonconvex in $\mathbf{x}$ but concave in $\mathbf{y}$. In general, there is no hope to find global optimum efficiently in nonconvex-concave setting.

More recently, nonconvex-nonconcave problem has gained more attention due to its generality. Particularly, there are several lines of work analyzing the dynamics of gradient descent-ascent (GDA) in nonconvex-nonconcave setting (such as GAN training). Though simple and intuitive, GDA has been shown to have undersirable convergence properties (Adolphs et al., 2019; Daskalakis and Panageas, 2018; Mazumdar et al., 2019; Jin et al., 2019) and exhibit strong rotation around fixed points (Mescheder et al., 2017; Balduzzi et al., 2018). To overcome this rotation behaviour of GDA, various modifications have been proposed, including averaging (Yazıcı et al., 2019), negative momentum (Gidel et al., 2019), extragradient (EG) (Korpelevich, 1976; Mertikopoulos et al., 2019), optimistic mirror descent (OGDA) (Daskalakis et al., 2018), consensus optimization (CO) (Mescheder et al., 2017) and symplectic gradient (SGA) (Balduzzi et al., 2018; Gemp and Mahadevan, 2018). However, we note that all these algorithms discard the underlying sequential structure of minimax optimization and adopt a simultaneous game formulation. In this work, we hold that GAN training is better viewed as a sequential game rather than a simultaneous game. The former is more consistent with the divergence minimization interpretation of GANs; there is also some empirical evidence showing that well-performing GAN generators are closer to a saddle-point instead of a local minimum (Berard et al., 2019), which suggests that local Nash, the typical solution concept for simultaneous games, may not be the most appropriate one for GANs.

To the best of our knowledge, the only two methods that can (and only) converge to local minimax are two time-scale GDA (Jin et al., 2019) and gradient dynamics with best response gradient (Fiez et al., 2019). In two time-scale GDA, the leader moves infinitely slower than the follower, which may cause slow convergence due to infinitely small learning rates. The dynamics in Fiez et al. (2019) is proposed for general-sum games. However, their main result for general-sum games require stronger assumptions and even in that case, the dynamics can converge to non-local-Stackelberg points in general-sum games. In contrast, in general-sum games, FR will not converge to non-local-Stackelberg points. Besides, Adolphs et al. (2019); Mazumdar et al. (2019) attempt to solve the undesirable convergence issue of GDA by exploiting curvature information, but they focus on simultaneous game on finding local Nash and it is unclear how to extend their algorithm to sequential games.

For GAN training, there is a rich literature on different strategies to make the GAN-game well-defined, *e.g.*, by adding instance noise (Salimans et al., 2016), by using different objectives (Nowozin et al., 2016; Gulrajani et al., 2017; Arjovsky et al., 2017; Mao et al., 2017) or by tweaking the architectures (Radford et al., 2015; Brock et al., 2019). While these strategies try to make the overall optimization problem easily, our work deals with a specific optimization problem and convergence issues arise in theory and in practice; hence our algorithm is orthogonal to these work.

## 6 EXPERIMENTS

In this section, we investigate whether the theoretical guarantees of FR carry over to practical problems. Particularly, our experiments have three main aims: (1) to test if FR converges and only converges to local minimax, (2) to test the effectiveness of FR in training GANs with saturating loss, (3) to test whether FR addresses the notorious rotation problem in GAN training.

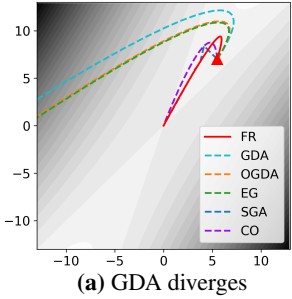 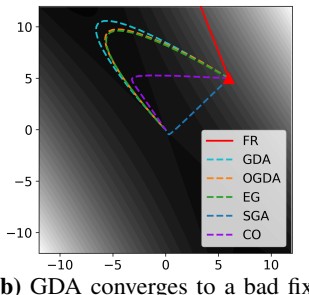 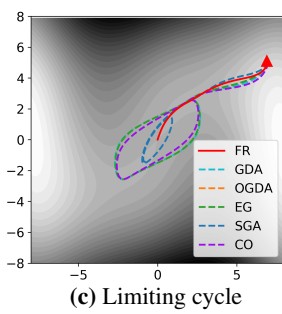

**(a)** GDA diverges      **(b)** GDA converges to a bad fixed point that is non local minimax      **(c)** Limiting cycle

**Figure 3:** Trajectory of FR and other algorithms in low dimensional toy problems. **Left:** for $g_1$, $(0,0)$ is local minimax. **Middle:** for $g_2$, $(0,0)$ is **NOT** local minimax. **Right:** for $g_3$, $(0,0)$ is a local minimax. The contours are for the function value. The red triangle marks the initial position.

## 6.1 LOW DIMENSIONAL TOY EXAMPLES

To verify our claim on exact local convergence, we first compare FR with gradient descent-ascent (GDA), optimistic mirror descent (OGDA) (Daskalakis et al., 2018), extragradient (EG) (Korpelevich, 1976), symplectic gradient adjustment (SGA) (Balduzzi et al., 2018) and consensus optimization (CO) (Mescheder et al., 2017) on three simple low dimensional problems:

$$g_1(x, y) = -3x^2 - y^2 + 4xy$$
$$g_2(x, y) = 3x^2 + y^2 + 4xy$$
$$g_3(x, y) = \left(4x^2 - (y - 3x + 0.05x^3)^2 - 0.1y^4\right) e^{-0.01(x^2+y^2)}.$$

Here $g_1$ and $g_2$ are two-dimensional quadratic problems, which are arguably the simplest nontrivial problems. $g_3$ is a sixth-order polynomial scaled by an exponential, which has a relatively complicated landscape compared to $g_1$ and $g_2$.

It can be seen that when running in $g_1$, where $(0,0)$ is a local (and in fact global) minimax, only FR, SGA and CO converge to it; all other method diverges (the trajectories of OGDA and EG almost overlap). The main reason behind the divergence of GDA is that gradient of leader pushes the system away from the local minimax when it is a local maximum for the leader. In $g_2$, where $(0,0)$ is not a local minimax, all algorithms except for FR converges to this undesired stationary point[4]. In this case, the leader is still to blame for the undesirable convergence of GDA (and other variants) since it gets trapped by the gradient pointing to the origin. In $g_3$, FR can converge to $(0,0)$, which is a local minimax, while all other methods apparently enter limit cycles around $(0,0)$. The experiments suggest that even on extremely simple instances, existing algorithms can either fail to converge to a desirable fixed point or converge to bad fixed points, whereas FR always exhibits desirable behaviors.

## 6.2 GENERATIVE ADVERSARIAL NETWORKS

One particularly promising application of minimax optimization algorithms is training generative adversarial networks (GANs). To recover the divergence minimization objective (e.g., Jensen-Shannon divergence in standard GANs), we have to model the adversarial game as a sequential game. According to the formulation, the generator is the leader who commits to an action first, while the discriminator is the follower that helps the generator to learn the target data distribution.

### 6.2.1 MIXTURE OF GAUSSIANS

We first evaluate 4 different algorithms (GDA, EG, CO and FR) on mixture of Gaussian problems with the original saturating loss. To satisfy the non-singular Hessian assumption, we add $L_2$ regularization (0.0002) to the discriminator. For both generator and discriminator, we use 2-hidden-layers MLP with 64 hidden units each layer where tanh activations is used. By default, RMSprop (Tieleman and Hinton, 2012) is used in all our experiments while the learning rate is tuned for GDA. As our FR involves the computation of Hessian inverses which is computational prohibitive, we instead use conjugate

---

[4]Note that it is a local *minimum* for the follower.

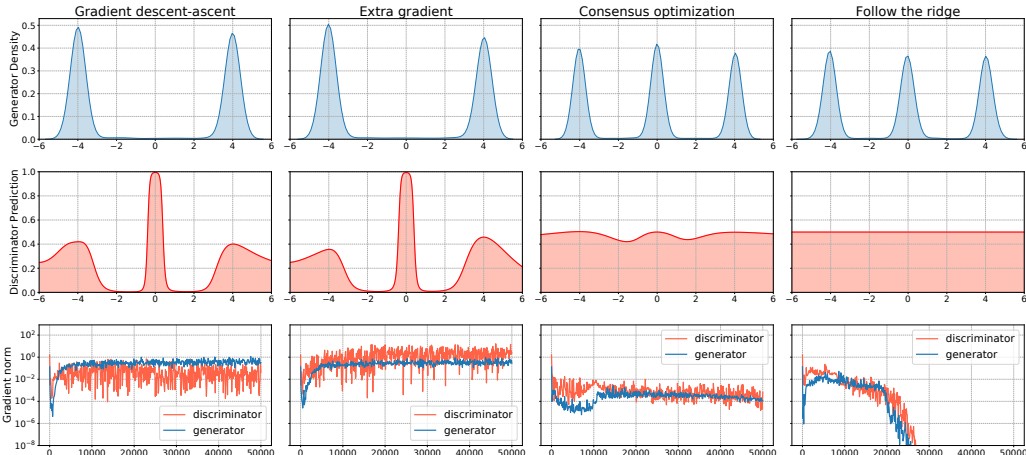

**Figure 4:** Comparison between FR and other algorithms on GANs with saturating loss. **First Row:** Generator distribution. Only consensus optimization (CO) and FR capture all three modes. **Second Row:** Discriminator prediction. The discriminator trained by FR converges to a flat line, indicating being fooled by the generator. **Third Row:** Gradient norm as a function of iteration. Only in the case of FR, the gradient norm vanishes.

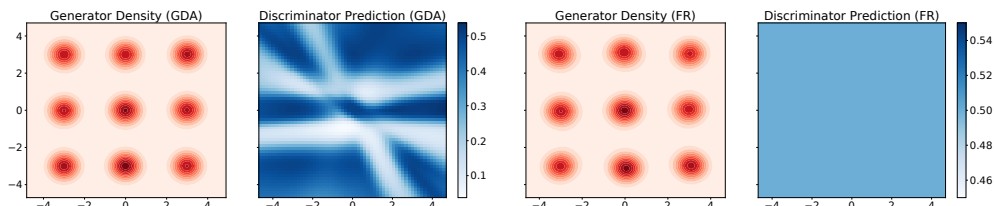

**Figure 5:** Comparison between FR and GDA on 2D mixture of Gaussians. **Left:** GDA; **Right:** FR.

gradient (Martens, 2010; Nocedal and Wright, 2006) to solve the linear system in the inner loop. To be specific, instead of solving $\mathbf{H_{yy}z} = \mathbf{H_{yx}}\nabla_\mathbf{x}f$ directly, we solve $\mathbf{H_{yy}^2 z} = \mathbf{H_{yy}H_{yx}}\nabla_\mathbf{x}f$ to ensure that the problem is well-posed since $\mathbf{H_{yy}^2}$ is always positive semidefinite. For all experimental details, we refer readers to Appendix D.2.

As shown in Fig. 4, GDA suffers from the "missing mode" problem and both discriminator and generator fail to converge as confirmed by the gradient norm plot. EG fails to resolve the convergence issue of GDA and performs similarly to GDA. With tuned gradient penalties, consensus optimization (CO) can successfully recover all three modes and obtain much smaller gradient norm. However, we notice that the gradient norm of CO decreases slowly and that both the generator and the discriminator have not converged after 50,000 iterations. In contrast, the generator

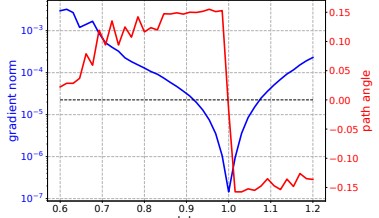

**Figure 6:** Path-norm and path-angle of FR along the linear path.

trained with FR successfully learns the true distribution with three modes and the discriminator is totally fooled by the generator. As expected, both players reach much lower gradient norm with FR, indicating fast convergence. Moreover, we find that even if initialized with GDA-trained networks (the top row of Fig. 4), FR can still find all the modes at the end of training.

To check whether FR fixes the strong rotation problem around fixed points, we follow Berard et al. (2019) to plot the gradient norm and path-angle (see Fig. 6). By interpolating between the initial parameters $\mathbf{z}$ and the final parameters $\mathbf{z}^*$, they proposed to monitor the angle between the vector field $\mathbf{v}$ and the linear path from $\mathbf{z}$ to $\mathbf{z}^*$. Specifically, they looked at the quantity – path-angle, defined as

$$\theta(\alpha) = \frac{\langle \mathbf{z}^* - \mathbf{z}, \mathbf{v}_\alpha \rangle}{\|\mathbf{z}^* - \mathbf{z}\|\|\mathbf{v}_\alpha\|} \text{ where } \mathbf{v}_\alpha = \mathbf{v}(\alpha\mathbf{z} + (1-\alpha)\mathbf{z}^*).$$

They showed that a high "bump" around $\alpha = 0$ in the path-angle plot typically indicates strong rotation behaviour. We choose $\alpha = [0.6, 1.2]$ and plot the gradient norm and path-angle along the linear path for the updates of FR. In particular, we only observe a sign-switch around the fixed point

$\mathbf{z}^*$ without an obvious bump, suggesting that FR doesn't exhibit rotational behaviour around the fixed point. To further check if FR converges to local minimax, we check the second-order condition of local minimax by computing the eigenvalues of $\mathbf{H_{xx}} - \mathbf{H_{xy}}\mathbf{H_{yy}^{-1}}\mathbf{H_{yx}}$ and $\mathbf{H_{yy}}$. As expected, all eigenvalues of $\mathbf{H_{xx}} - \mathbf{H_{xy}}\mathbf{H_{yy}^{-1}}\mathbf{H_{yx}}$ are non-negative while all eigenvalues of $\mathbf{H_{yy}}$ are non-positive.

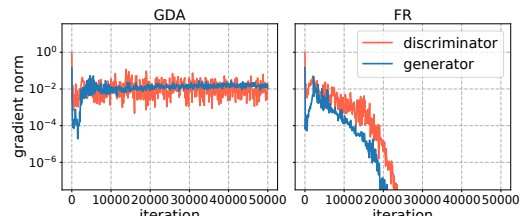

We also run FR on 2-D mixture of Gaussian with the same architectures (see Fig. 5) and compare it to vanilla GDA. Though GDA captures all the modes, we note that both the generator and the discriminator don't converge which can be seen from the gradient norm plot in Fig. 12. In contrast, the discriminator trained by FR is totally fooled by the generator and gradients vanish. We stress here that the sample quality in GAN

**Figure 7:** Gradient norms of GDA and FR.

models is not a good metric of checking convergence as we shown in the above example.

### 6.2.2 PRELIMINARY RESULTS ON MNIST

In a more realistic setting, we test our algorithm on image generation task. Particularly, we use the standard MNIST dataset (LeCun et al., 1998) but only take a subset of the dataset with class 0 and 1 for quick experimenting. To stabilize the training of GANs, we employ spectral normalization (Miyato et al., 2018) to enforce Lipschitz continuity on the discriminator. To ensure the invertibility of the discriminator's Hessian, we add the same amount of $L_2$ regularization to the discriminator as in mixture of Gaussian experiments. In terms of network architectures, we use 2-hidden-layers MLP with 512 hidden units in each layer for both the discriminator and the generator. For the discriminator, we use Sigmoid activation in the output layer. We use RMSProp as our base optimizer in the experiments with batch size 2,000. We run both GDA and FR for 100,000 iterations.

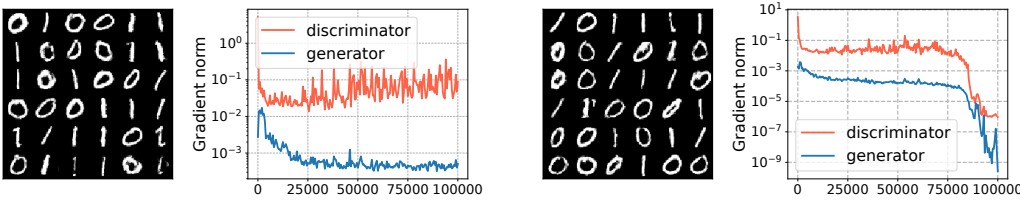

**Figure 8:** Comparison between FR and GDA on MNIST dataset. **Left:** GDA; **Right:** FR.

In Fig. 8, we show the generated samples of GDA and FR along with the gradient norm plots. Our main observation is that FR improves convergence as the gradient norms of both discriminator and generator decrease much faster than GDA; however the convergence is not well reflected by the quality of generated samples. We notice that gradients don't vanish to zero at the end of training. We conjecture that for high-dimensional data distribution like images, the network we used is not flexible enough to learn the distribution perfectly.

## 7 CONCLUSION

In this paper, we studied local convergence of learning dynamics in minimax optimization. To address undesirable behaviours of gradient descent-ascent, we proposed a novel algorithm that locally converges to and only converges to local minimax by taking into account the sequential structure of minimax optimization. Meanwhile, we proved that our algorithm addresses the notorious rotational behaviour of vanilla gradient-descent-ascent around fixed points. We further showed theoretically that our algorithm is compatible with standard acceleration techniques, including preconditioning and positive momentum. Our algorithm can be easily extended to general-sum Stackelberg games with similar theoretical guarantees. Empirically, we validated the effectiveness of our algorithm in both low-dimensional toy problems and GAN training.

## ACKNOWLEDGEMENT

We thank Kefan Dong, Roger Grosse and Shengyang Sun for helpful comments on this project.

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

## A    PROOF OF THEOREM 1

*Proof.* First of all, note that FR's update rule can be rewritten as

$$\begin{bmatrix} \mathbf{x}_{t+1} \\ \mathbf{y}_{t+1} \end{bmatrix} \leftarrow \begin{bmatrix} \mathbf{x}_t \\ \mathbf{y}_t \end{bmatrix} - \eta_{\mathbf{x}} \begin{bmatrix} \mathbf{I} & \\ -\mathbf{H}_{\mathbf{yy}}^{-1}\mathbf{H}_{\mathbf{yx}} & c\mathbf{I} \end{bmatrix} \begin{bmatrix} \nabla_{\mathbf{x}} f \\ -\nabla_{\mathbf{y}} f \end{bmatrix}, \tag{5}$$

where $c := \eta_{\mathbf{y}}/\eta_{\mathbf{x}}$, and that $\begin{bmatrix} \mathbf{I} & \\ -\mathbf{H}_{\mathbf{yy}}^{-1}\mathbf{H}_{\mathbf{yx}} & c\mathbf{I} \end{bmatrix}$ is always invertible. Therefore, the fixed points of FR are exactly those that satisfy $\nabla f(\mathbf{x}, \mathbf{y}) = 0$, *i.e.*, the first-order necessary condition of local minimax. Now, consider a fixed point $(\mathbf{x}^*, \mathbf{y}^*)$. The Jacobian of FR's update rule at $(\mathbf{x}^*, \mathbf{y}^*)$ is given by

$$\mathbf{J} = \mathbf{I} - \eta_{\mathbf{x}} \begin{bmatrix} \mathbf{I} & \\ -\mathbf{H}_{\mathbf{yy}}^{-1}\mathbf{H}_{\mathbf{yx}} & \mathbf{I} \end{bmatrix} \begin{bmatrix} \mathbf{H}_{\mathbf{xx}} & \mathbf{H}_{\mathbf{xy}} \\ -c\mathbf{H}_{\mathbf{yx}} & -c\mathbf{H}_{\mathbf{yy}} \end{bmatrix}.$$

Observe that $\mathbf{J}$ is similar to

$$\begin{bmatrix} \mathbf{I} & \\ \mathbf{H}_{\mathbf{yy}}^{-1}\mathbf{H}_{\mathbf{yx}} & \mathbf{I} \end{bmatrix} \mathbf{J} \begin{bmatrix} \mathbf{I} & \\ -\mathbf{H}_{\mathbf{yy}}^{-1}\mathbf{H}_{\mathbf{yx}} & \mathbf{I} \end{bmatrix}$$

$$= \mathbf{I} - \eta_{\mathbf{x}} \begin{bmatrix} \mathbf{I} & \\ \mathbf{H}_{\mathbf{yy}}^{-1}\mathbf{H}_{\mathbf{yx}} & \mathbf{I} \end{bmatrix} \begin{bmatrix} \mathbf{I} & \\ -\mathbf{H}_{\mathbf{yy}}^{-1}\mathbf{H}_{\mathbf{yx}} & \mathbf{I} \end{bmatrix} \begin{bmatrix} \mathbf{H}_{\mathbf{xx}} & \mathbf{H}_{\mathbf{xy}} \\ -c\mathbf{H}_{\mathbf{yx}} & -c\mathbf{H}_{\mathbf{yy}} \end{bmatrix} \begin{bmatrix} \mathbf{I} & \\ -\mathbf{H}_{\mathbf{yy}}^{-1}\mathbf{H}_{\mathbf{yx}} & \mathbf{I} \end{bmatrix}$$

$$= \mathbf{I} - \eta_{\mathbf{x}} \begin{bmatrix} \mathbf{H}_{\mathbf{xx}} - \mathbf{H}_{\mathbf{xy}}\mathbf{H}_{\mathbf{yy}}^{-1}\mathbf{H}_{\mathbf{yx}} & \mathbf{H}_{\mathbf{xy}} \\ & -c\mathbf{H}_{\mathbf{yy}} \end{bmatrix},$$

which is block diagonal. Therefore, the eigenvalues of $\mathbf{J}$ are exactly those of $\mathbf{I} + \eta_{\mathbf{y}}\mathbf{H}_{\mathbf{yy}}$ and those of $\mathbf{I} - \eta_{\mathbf{x}}(\mathbf{H}_{\mathbf{xx}} - \mathbf{H}_{\mathbf{xy}}\mathbf{H}_{\mathbf{yy}}^{-1}\mathbf{H}_{\mathbf{yx}})$, which are all real because both matrices are symmetric.

Moreover, suppose that

$$\eta_{\mathbf{x}} < \frac{2}{\max\left\{\rho(\mathbf{H}_{\mathbf{xx}} - \mathbf{H}_{\mathbf{xy}}\mathbf{H}_{\mathbf{yy}}^{-1}\mathbf{H}_{\mathbf{yx}}), c\rho(-\mathbf{H}_{\mathbf{yy}})\right\}},$$

where $\rho(\cdot)$ stands for spectral radius. In this case

$$-\mathbf{I} \prec \mathbf{I} + \eta_{\mathbf{y}}\mathbf{H}_{\mathbf{yy}}, \quad -\mathbf{I} \prec \mathbf{I} - \eta_{\mathbf{x}}(\mathbf{H}_{\mathbf{xx}} - \mathbf{H}_{\mathbf{xy}}\mathbf{H}_{\mathbf{yy}}^{-1}\mathbf{H}_{\mathbf{yx}}).$$

Therefore whether $\rho(\mathbf{J}) < 1$ depends on whether $-\mathbf{H}_{\mathbf{yy}}$ or $\mathbf{H}_{\mathbf{xx}} - \mathbf{H}_{\mathbf{xy}}\mathbf{H}_{\mathbf{yy}}^{-1}\mathbf{H}_{\mathbf{yx}}$ has negative eigenvalues. If $(\mathbf{x}^*, \mathbf{y}^*)$ is a local minimax, by the necessary condition, $\mathbf{H}_{\mathbf{yy}} \preccurlyeq \mathbf{0}$, $\mathbf{H}_{\mathbf{xx}} - \mathbf{H}_{\mathbf{xy}}\mathbf{H}_{\mathbf{yy}}^{-1}\mathbf{H}_{\mathbf{yx}} \succcurlyeq \mathbf{0}$. It follows that the eigenvalues of $\mathbf{J}$ all fall in $(-1, 1]$. $(\mathbf{x}^*, \mathbf{y}^*)$ is thus a stable fixed point of FR.

On the other hand, when $(\mathbf{x}^*, \mathbf{y}^*)$ is a strictly stable fixed point, $\rho(\mathbf{J}) < 1$. It follows that both $\mathbf{H}_{\mathbf{yy}}$ and $\mathbf{H}_{\mathbf{xx}} - \mathbf{H}_{\mathbf{xy}}\mathbf{H}_{\mathbf{yy}}^{-1}\mathbf{H}_{\mathbf{yx}}$ must be positive definite. By the sufficient conditions of local minimax, $(\mathbf{x}^*, \mathbf{y}^*)$ is a local minimax. $\qquad\square$

## B    PROOF OF THEOREM 2

Consider a general discrete dynamical system $\mathbf{z}_{t+1} \leftarrow g(\mathbf{z}_t)$. Let $\mathbf{z}^*$ be a fixed point of $g(\cdot)$. Let $\mathbf{J}(\mathbf{z})$ denote the Jacobian of $g(\cdot)$ at $\mathbf{z}$. Similar results can be found in many texts; see, for instance, Theorem 2.12 (Olver, 2015).

**Proposition 4** (Local convergence rate from Jacobian eigenvalue). *If $\rho(\mathbf{J}(\mathbf{z}^*)) = 1 - \Delta < 1$, then there exists a neighborhood $U$ of $\mathbf{z}^*$ such that for any $\mathbf{z}_0 \in U$,*

$$\|\mathbf{z}_t - \mathbf{z}^*\|_2 \le C\left(1 - \frac{\Delta}{2}\right)^t \|\mathbf{z}_0 - \mathbf{z}^*\|_2,$$

*where $C$ is some constant.*

*Proof.* By Lemma 5.6.10 (Horn and Johnson, 2013), since $\rho(\mathbf{J}(\mathbf{z}^*)) = 1 - \Delta$, there exists a matrix norm $\|\cdot\|$ induced by vector norm $\|\cdot\|$ such that $\|\mathbf{J}(\mathbf{z}^*)\| < 1 - \frac{3\Delta}{4}$. Now consider the Taylor expansion of $g(\mathbf{z})$ at the fixed point $\mathbf{z}^*$:

$$g(\mathbf{z}) = g(\mathbf{z}^*) + \mathbf{J}(\mathbf{z}^*)(\mathbf{z} - \mathbf{z}^*) + R(\mathbf{z} - \mathbf{z}^*),$$

where the remainder term satisfies

$$\lim_{\mathbf{z}\to\mathbf{z}^*} \frac{R(\mathbf{z}-\mathbf{z}^*)}{\|\mathbf{z}-\mathbf{z}^*\|} = 0.$$

Therefore, we can choose $0 < \delta$ such that whenever $\|\mathbf{z}-\mathbf{z}^*\| < \delta$, $\|R(\mathbf{z}-\mathbf{z}^*)\| \le \frac{\Delta}{4}\|\mathbf{z}-\mathbf{z}^*\|$. In this case,

$$\|g(\mathbf{z}) - g(\mathbf{z}^*)\| \le \|\mathbf{J}(\mathbf{z}^*)(\mathbf{z}-\mathbf{z}^*)\| + \|R(\mathbf{z}-\mathbf{z}^*)\|$$

$$\le \|\mathbf{J}(\mathbf{z}^*)\|\|\mathbf{z}-\mathbf{z}^*\| + \frac{\Delta}{4}\|\mathbf{z}-\mathbf{z}^*\|$$

$$\le \left(1 - \frac{\Delta}{2}\right)\|\mathbf{z}-\mathbf{z}^*\|.$$

In other words, when $\mathbf{z}_0 \in U = \{\mathbf{z} |\ \|\mathbf{z}-\mathbf{z}^*\| < \delta\}$,

$$\|\mathbf{z}_t - \mathbf{z}^*\| \le \left(1 - \frac{\Delta}{2}\right)^t \|\mathbf{z}_0 - \mathbf{z}^*\|.$$

By the equivalence of finite dimensional norms, there exists constants $c_1, c_2 > 0$ such that

$$\forall \mathbf{z}, \quad c_1\|\mathbf{z}\|_2 \le \|\mathbf{z}\| \le c_2\|\mathbf{z}\|_2.$$

Therefore

$$\|\mathbf{z}_t - \mathbf{z}^*\|_2 \le \frac{c_2}{c_1}\left(1 - \frac{\Delta}{2}\right)^t \|\mathbf{z}_0 - \mathbf{z}^*\|_2.$$

$\square$

In other words, the rate of convergence is given by the gap between $\rho(\mathbf{J})$ and 1. We now prove Theorem 2 using this view.

*proof of Theorem 2.* In the following proof we use $\|\cdot\|$ to denote the standard spectral norm. It is not hard to see that $\lambda_{max}(-\mathbf{H_{yy}}) \le \rho(\nabla^2 f(\mathbf{x}^*, \mathbf{y}^*)) = \beta$ and $\|\mathbf{H_{xy}}\| \le \beta$. Also,

$$\lambda_{max}(\mathbf{H_{xx}} - \mathbf{H_{xy}}\mathbf{H_{yy}^{-1}}\mathbf{H_{yx}}) \le \|\mathbf{H_{xx}}\| + \|\mathbf{H_{xy}}\|^2 \cdot \|\mathbf{H_{yy}^{-1}}\| \le \beta + \frac{\beta^2}{\alpha} = (1+\kappa)\beta.$$

Therefore we choose our learning rate to be $\eta_{\mathbf{x}} = \eta_{\mathbf{y}} = \frac{1}{2\kappa\beta}$. In this case, the eigenvalues of the Jacobian of FR without momentum all fall in $\left[0, 1 - \frac{1}{2\kappa^2}\right]$. Using Proposition 4, we can show that FR locally converges with a rate of $\Omega(\kappa^{-2})$.

Now, let us consider FR with momentum:

$$\begin{bmatrix} \mathbf{x}_{t+1} \\ \mathbf{y}_{t+1} \end{bmatrix} \leftarrow \begin{bmatrix} \mathbf{x}_t \\ \mathbf{y}_t \end{bmatrix} - \eta_{\mathbf{x}} \begin{bmatrix} \mathbf{I} & \\ -\mathbf{H_{yy}^{-1}}\mathbf{H_{yx}} & \mathbf{I} \end{bmatrix} \begin{bmatrix} \nabla_{\mathbf{x}} f \\ -\nabla_{\mathbf{y}} f \end{bmatrix} + \gamma \begin{bmatrix} \mathbf{x}_t - \mathbf{x}_{t-1} \\ \mathbf{y}_t - \mathbf{y}_{t-1} \end{bmatrix}. \tag{6}$$

This is a dynamical system on the augmented space of $(\mathbf{x}_t, \mathbf{y}_t, \mathbf{x}_{t-1}, \mathbf{y}_{t-1})$. Let

$$\mathbf{J}_1 := \mathbf{I} - \eta_{\mathbf{x}} \begin{bmatrix} \mathbf{I} & \\ -\mathbf{H_{yy}^{-1}}\mathbf{H_{yx}} & \mathbf{I} \end{bmatrix} \begin{bmatrix} \mathbf{H_{xx}} & \mathbf{H_{xy}} \\ -\mathbf{H_{yx}} & -\mathbf{H_{yy}} \end{bmatrix}$$

be the Jacobian of the original FR at a fixed point $(\mathbf{x}^*, \mathbf{y}^*)$. Then the Jacobian of Polyak's momentum at $(\mathbf{x}^*, \mathbf{y}^*, \mathbf{x}^*, \mathbf{y}^*)$ is

$$\mathbf{J}_2 := \begin{bmatrix} \gamma\mathbf{I} + \mathbf{J}_1 & -\gamma\mathbf{I} \\ \mathbf{I} & 0 \end{bmatrix}.$$

The spectrum of $\mathbf{J}_2$ is given by solutions to

$$\det(\lambda\mathbf{I} - \mathbf{J}_2) = \det\left((\lambda^2 - \gamma\lambda + \gamma)\mathbf{I} - \gamma\mathbf{J}_1\right) = 0.$$

In other words, an eigenvalue $r$ of $\mathbf{J}_1$ corresponds to two eigenvalues of $\mathbf{J}_2$ given by the roots of $\lambda^2 - (\gamma+r)\lambda + \gamma = 0$. For our case, let us choose $\gamma = 1 + \frac{1}{2\kappa^2} - \frac{\sqrt{2}}{\kappa}$. Then for any $r \in \left[0, 1 - \frac{1}{2\kappa^2}\right]$,

$$(r+\gamma)^2 - 4\gamma \le \left(1 - \frac{1}{2\kappa^2} + \gamma\right)^2 - 4\gamma = 0.$$

Therefore the two roots of $\lambda^2 - (\gamma+r)\lambda + \gamma = 0$ must be imaginary, and their magnitude are exactly $\sqrt{\gamma}$. Since $\sqrt{\gamma} \le 1 - \frac{1-\gamma}{2} \le 1 - \frac{1}{2\sqrt{2}\kappa}$, we now know that $\rho(\mathbf{J}_2) \le 1 - \frac{1}{2\sqrt{2}\kappa}$. Using Proposition 4, we can see that FR with momentum locally converge with a rate of $\Omega(\kappa^{-1})$. $\square$

## C  PROOFS FOR SECTION 4

### C.1  PRECONDITIONING

Recall that the preconditioned variant of FR is given by

$$
\begin{bmatrix} \mathbf{x}_{t+1} \\ \mathbf{y}_{t+1} \end{bmatrix} \leftarrow \begin{bmatrix} \mathbf{x}_t \\ \mathbf{y}_t \end{bmatrix} - \begin{bmatrix} \mathbf{I} & \\ -\mathbf{H}_{\mathbf{yy}}^{-1}\mathbf{H}_{\mathbf{yx}} & \mathbf{I} \end{bmatrix} \begin{bmatrix} \eta_{\mathbf{x}}\mathbf{P}_1\nabla_{\mathbf{x}}f \\ -\eta_{\mathbf{y}}\mathbf{P}_2\nabla_{\mathbf{y}}f \end{bmatrix}. \tag{7}
$$

We now prove that preconditioning does not effect the local convergence properties.

**Proposition 5.** *If $A$ is a symmetric real matrix, $B$ is symmetric and positive definite, then the eigenvalues of $AB$ are all real, and $AB$ and $A$ have the same number of positive, negative and zero eigenvalues.*

*Proof.* $AB$ is similar to and thus has the same eigenvalues as $B^{\frac{1}{2}}AB^{\frac{1}{2}}$, which is symmetric and has real eigenvalues. Since $B^{\frac{1}{2}}AB^{\frac{1}{2}}$ is congruent to $A$, they have the same number of positive, negative and zero eigenvalues (see Horn and Johnson (2013, Theorem 4.5.8)). □

**Proposition 6.** *Assume that $\mathbf{P}_1$ and $\mathbf{P}_2$ are positive definite. The Jacobian of (7) has only real eigenvalues at fixed points. With a suitable learning rate, all strictly stable fixed points of (7) are local minimax, and all local minimax are stable fixed points of (7).*

*Proof.* First, observe that both $\begin{bmatrix} \mathbf{I} & \\ -\mathbf{H}_{\mathbf{yy}}^{-1}\mathbf{H}_{\mathbf{yx}} & \mathbf{I} \end{bmatrix}$ and $\begin{bmatrix} \mathbf{P}_1 & \\ & \mathbf{P}_2 \end{bmatrix}$ are both always invertible. Hence fixed points of (7) are exactly stationary points. Let $c := \eta_{\mathbf{y}}/\eta_{\mathbf{x}}$. Note that the Jacobian of (7) is given by

$$
\mathbf{J} = \mathbf{I} - \eta_{\mathbf{x}} \begin{bmatrix} \mathbf{I} & \\ -\mathbf{H}_{\mathbf{yy}}^{-1}\mathbf{H}_{\mathbf{yx}} & \mathbf{I} \end{bmatrix} \begin{bmatrix} \mathbf{P}_1 & \\ & \mathbf{P}_2 \end{bmatrix} \begin{bmatrix} \mathbf{H}_{\mathbf{xx}} & \mathbf{H}_{\mathbf{xy}} \\ -c\mathbf{H}_{\mathbf{yx}} & -c\mathbf{H}_{\mathbf{yy}} \end{bmatrix},
$$

which is similar to

$$
\begin{bmatrix} \mathbf{I} & \\ \mathbf{H}_{\mathbf{yy}}^{-1}\mathbf{H}_{\mathbf{yx}} & \mathbf{I} \end{bmatrix} \mathbf{J} \begin{bmatrix} \mathbf{I} & \\ -\mathbf{H}_{\mathbf{yy}}^{-1}\mathbf{H}_{\mathbf{yx}} & \mathbf{I} \end{bmatrix}
$$
$$
= \mathbf{I} - \eta_{\mathbf{x}} \begin{bmatrix} \mathbf{P}_1 & \\ & \mathbf{P}_2 \end{bmatrix} \begin{bmatrix} \mathbf{H}_{\mathbf{xx}} - \mathbf{H}_{\mathbf{xy}}\mathbf{H}_{\mathbf{yy}}^{-1}\mathbf{H}_{\mathbf{yx}} & \mathbf{H}_{\mathbf{xy}} \\ & -c\mathbf{H}_{\mathbf{yy}} \end{bmatrix}.
$$

Therefore the eigenvalues of $\mathbf{J}$ are exactly those of $\mathbf{I} - \eta_{\mathbf{x}}\mathbf{P}_1\left(\mathbf{H}_{\mathbf{xx}} - \mathbf{H}_{\mathbf{xy}}\mathbf{H}_{\mathbf{yy}}^{-1}\mathbf{H}_{\mathbf{yx}}\right)$ and $\mathbf{I} + \eta_{\mathbf{y}}\mathbf{P}_2\mathbf{H}_{\mathbf{yy}}$. By Proposition 5, the eigenvalues of both matrices are all real. When the learning rates are small enough, *i.e.*, when

$$
\eta_{\mathbf{x}} < \frac{2}{\max\left\{\rho\left(\mathbf{P}_1(\mathbf{H}_{\mathbf{xx}} - \mathbf{H}_{\mathbf{xy}}\mathbf{H}_{\mathbf{yy}}^{-1}\mathbf{H}_{\mathbf{yx}})\right), c\rho(-\mathbf{P}_2\mathbf{H}_{\mathbf{yy}})\right\}},
$$

whether $\rho(\mathbf{J}) \leq 1$ solely depends on whether $\mathbf{P}_1\left(\mathbf{H}_{\mathbf{xx}} - \mathbf{H}_{\mathbf{xy}}\mathbf{H}_{\mathbf{yy}}^{-1}\mathbf{H}_{\mathbf{yx}}\right)$ and $-\mathbf{P}_2\mathbf{H}_{\mathbf{yy}}$ have negative eigenvalues. By Proposition 5, the number of positive, negative and zero eigenvalues of the two matrices are the same as those of $\mathbf{H}_{\mathbf{xx}} - \mathbf{H}_{\mathbf{xy}}\mathbf{H}_{\mathbf{yy}}^{-1}\mathbf{H}_{\mathbf{yx}}$ and $-\mathbf{H}_{\mathbf{yy}}$ respectively. Therefore the proposition follows from the same argument as in Theorem 1. □

### C.2  GENERAL-SUM STACKELBERG GAMES

A general-sum Stackelberg game is formulated as follows. There is a leader, whose action is $\mathbf{x} \in \mathbb{R}^n$, and a follower, whose action is $\mathbf{y} \in \mathbb{R}^m$. The leader's cost function is given by $f(\mathbf{x}, \mathbf{y})$ while the follower's is given by $g(\mathbf{x}, \mathbf{y})$. The generalization of minimax in general-sum Stackelberg games is *Stackelberg equilibrium*.

**Definition 3** (Stackelberg equilibrium). *$(\mathbf{x}^*, \mathbf{y}^*)$ is a (global) Stackelberg equilibrium if $\mathbf{y}^* \in R(\mathbf{x}^*)$, and $\forall \mathbf{x} \in \mathcal{X}$,*

$$
f(\mathbf{x}^*, \mathbf{y}^*) \leq \max_{\mathbf{y} \in R(\mathbf{x})} f(\mathbf{x}, \mathbf{y}),
$$

*where $R(\mathbf{x}) := \arg\min g(\mathbf{x}, \cdot)$ is the best response set for the follower.*

Similarly, we have local Stackelberg equilibrium (Fiez et al., 2019) defined as follows.[5]

**Definition 4** (Local Stackelberg equilibrium). $(\mathbf{x}^*, \mathbf{y}^*)$ *is a local Stackelberg equilibrium if*

1. $\mathbf{y}^*$ *is a local minimum of* $g(\mathbf{x}^*, \cdot)$;

2. *Let* $r(\mathbf{x})$ *be the implicit function defined by* $\nabla_{\mathbf{y}} g(\mathbf{x}, \mathbf{y}) = 0$ *in a neighborhood of* $\mathbf{x}^*$ *with* $r(\mathbf{x}^*) = \mathbf{y}^*$. *Then* $\mathbf{x}^*$ *is a local minimum of* $\phi(\mathbf{x}) := f(\mathbf{x}, r(\mathbf{x}))$.

For local Stackelberg equilibrium, we have similar necessary conditions and sufficient conditions. For simplicity, we use the following notation when it is clear from the context

$$\nabla^2 f(\mathbf{x}, \mathbf{y}) = \begin{bmatrix} \mathbf{H_{xx}} & \mathbf{H_{xy}} \\ \mathbf{H_{yx}} & \mathbf{H_{yy}} \end{bmatrix}, \ \nabla^2 g(\mathbf{x}, \mathbf{y}) = \begin{bmatrix} \mathbf{G_{xx}} & \mathbf{G_{xy}} \\ \mathbf{G_{yx}} & \mathbf{G_{yy}} \end{bmatrix}.$$

Similar to the zero-sum case, local Stackelberg equilibrium can be characterized using derivatives.

**Proposition 7** (Necessary conditions). *Any local Stackelberg equilibrium satisfies* $\nabla_{\mathbf{y}} g(\mathbf{x}, \mathbf{y}) = 0$, $\nabla_{\mathbf{x}} f(\mathbf{x}, \mathbf{y}) - \mathbf{G_{xy}} \mathbf{G_{yy}^{-1}} \nabla_{\mathbf{y}} f(\mathbf{x}, \mathbf{y}) = 0$, $\nabla^2_{\mathbf{yy}} g(\mathbf{x}, \mathbf{y}) \succcurlyeq 0$ *and*

$$\mathbf{H_{xx}} - \mathbf{H_{xy}} \mathbf{G_{yy}^{-1}} \mathbf{G_{yx}} - \nabla_{\mathbf{x}} \left( \mathbf{G_{xy}} \mathbf{G_{yy}^{-1}} \nabla_{\mathbf{y}} f \right) + \nabla_{\mathbf{y}} \left( \mathbf{G_{xy}} \mathbf{G_{yy}^{-1}} \nabla_{\mathbf{y}} f \right) \mathbf{G_{yy}^{-1}} \mathbf{G_{yx}} \succcurlyeq 0.$$

**Proposition 8** (Sufficient conditions). *If* $(\mathbf{x}, \mathbf{y})$ *satisfy* $\nabla_{\mathbf{y}} g(\mathbf{x}, \mathbf{y}) = 0$, $\nabla_{\mathbf{x}} f(\mathbf{x}, \mathbf{y}) - \mathbf{G_{xy}} \mathbf{G_{yy}^{-1}} \nabla_{\mathbf{y}} f(\mathbf{x}, \mathbf{y}) = 0$, $\nabla^2_{\mathbf{yy}} g(\mathbf{x}, \mathbf{y}) \succ 0$ *and*

$$\mathbf{H_{xx}} - \mathbf{H_{xy}} \mathbf{G_{yy}^{-1}} \mathbf{G_{yx}} - \nabla_{\mathbf{x}} \left( \mathbf{G_{xy}} \mathbf{G_{yy}^{-1}} \nabla_{\mathbf{y}} f \right) + \nabla_{\mathbf{y}} \left( \mathbf{G_{xy}} \mathbf{G_{yy}^{-1}} \nabla_{\mathbf{y}} f \right) \mathbf{G_{yy}^{-1}} \mathbf{G_{yx}} \succ 0.$$

*then* $(\mathbf{x}, \mathbf{y})$ *is a local Stackelberg equilibrium.*

The conditions above can be derived from the definition with the observation that

$$\begin{aligned} \nabla^2 \phi(\mathbf{x}) &= \nabla \left( \nabla_{\mathbf{x}} f(\mathbf{x}, r(\mathbf{x}))^\top - \nabla_{\mathbf{y}} f(\mathbf{x}, r(\mathbf{x}))^\top \nabla r(\mathbf{x}) \right) \\ &= \nabla^2_{\mathbf{xx}} f + \nabla^2_{\mathbf{xy}} f \nabla r(\mathbf{x}) + \nabla_{\mathbf{x}} \left( \nabla_{\mathbf{y}} f^\top \nabla r(\mathbf{x}) \right) + \nabla_{\mathbf{y}} \left( \nabla_{\mathbf{y}} f^\top \nabla r(\mathbf{x}) \right) \nabla r(\mathbf{x}) \\ &= \mathbf{H_{xx}} - \mathbf{H_{xy}} \mathbf{G_{yy}^{-1}} \mathbf{G_{yx}} - \nabla_{\mathbf{x}} \left( \mathbf{G_{xy}} \mathbf{G_{yy}^{-1}} \nabla_{\mathbf{y}} f \right) + \nabla_{\mathbf{y}} \left( \mathbf{G_{xy}} \mathbf{G_{yy}^{-1}} \nabla_{\mathbf{y}} f \right) \mathbf{G_{yy}^{-1}} \mathbf{G_{yx}}. \end{aligned}$$

Here all derivatives are evaluated at $(\mathbf{x}, r(\mathbf{x}))$. We would like to clarify that by $\nabla_{\mathbf{x}} h$, where $h : \mathbb{R}^{n+m} \to \mathbb{R}$, we mean the partial derivative of $h$ for the first $n$ entries. Similarly for $h : \mathbb{R}^{n+m} \to \mathbb{R}^k$, by $\nabla_{\mathbf{x}} h$ we mean the first $n$ columns of the Jacobian of $h$, which is $k$-by-$(n+m)$.

Henceforth we will use $D_{\mathbf{x}} f(\mathbf{x}, \mathbf{y})$ to denote $\nabla_{\mathbf{x}} f - \mathbf{G_{xy}} \mathbf{G_{yy}^{-1}} \nabla_{\mathbf{y}} f(\mathbf{x}, \mathbf{y})$. The general-sum version of Follow-the-Ridge is given by

$$\begin{bmatrix} \mathbf{x}_{t+1} \\ \mathbf{y}_{t+1} \end{bmatrix} \leftarrow \begin{bmatrix} \mathbf{x}_t \\ \mathbf{y}_t \end{bmatrix} - \begin{bmatrix} \mathbf{I} & \\ -\mathbf{G_{yy}^{-1}} \mathbf{G_{yx}} & \mathbf{I} \end{bmatrix} \begin{bmatrix} \eta_{\mathbf{x}} D_{\mathbf{x}} f(\mathbf{x}_t, \mathbf{y}_t) \\ \eta_{\mathbf{y}} \nabla_{\mathbf{y}} g(\mathbf{x}_t, \mathbf{y}_t) \end{bmatrix}. \tag{8}$$

Just as the zero-sum version of FR converges exactly to local minimax, we can show that the general-sum version of FR converges exactly to local Stackelberg equilibria. As in the zero-sum setting, we use stability of fixed points as a proxy of discussing local convergence.

**Theorem 3.** *The Jacobian of (8) has only real eigenvalues at fixed points. With a suitable learning rate, all strictly stable fixed points of (8) are local Stackelberg equilibria, and all local Stackelberg equilibria are stable fixed points of (8).*

*Proof.* This theorem only analyzes the Jacobian of (8) **at fixed points**; thus we will only need to focus on the fixed points of (8) in the proof.

Let $c := \eta_{\mathbf{y}} / \eta_{\mathbf{x}}$. Note that $\begin{bmatrix} \mathbf{I} & \\ -\mathbf{G_{yy}^{-1}} \mathbf{G_{yx}} & \mathbf{I} \end{bmatrix}$ is always invertible. Therefore, the fixed points of (8) are exactly points $(\mathbf{x}, \mathbf{y})$ that satisfy $D_{\mathbf{x}} f(\mathbf{x}, \mathbf{y}) = 0$ and $\nabla_{\mathbf{y}} g(\mathbf{x}, \mathbf{y}) = 0$, i.e. the first-order necessary condition for local Stackelberg equilibria.

---

[5]Our definition is slightly different from that in Fiez et al. (2019)

In particular, consider a **fixed point** $\mathbf{z}^* := (\mathbf{x}, \mathbf{y})$. The Jacobian of (8) at $(\mathbf{x}, \mathbf{y})$ is given by

$$\mathbf{J} = \mathbf{I} - \eta_{\mathbf{x}} \begin{bmatrix} \mathbf{I} & \mathbf{I} \\ -\mathbf{G}_{\mathbf{yy}}^{-1}\mathbf{G}_{\mathbf{yx}} & \mathbf{I} \end{bmatrix} \begin{bmatrix} \mathbf{H}_{\mathbf{xx}} - \nabla_{\mathbf{x}}(\mathbf{G}_{\mathbf{xy}}\mathbf{G}_{\mathbf{yy}}^{-1}\nabla_{\mathbf{y}}f) & \mathbf{H}_{\mathbf{xy}} - \nabla_{\mathbf{y}}(\mathbf{G}_{\mathbf{xy}}\mathbf{G}_{\mathbf{yy}}^{-1}\nabla_{\mathbf{y}}f) \\ c\mathbf{G}_{\mathbf{yx}} & c\mathbf{G}_{\mathbf{yy}} \end{bmatrix}.$$

Observe that

$$\begin{bmatrix} \mathbf{I} & \\ \mathbf{G}_{\mathbf{yy}}^{-1}\mathbf{G}_{\mathbf{yx}} & \mathbf{I} \end{bmatrix} \mathbf{J} \begin{bmatrix} \mathbf{I} & \\ -\mathbf{G}_{\mathbf{yy}}^{-1}\mathbf{G}_{\mathbf{yx}} & \mathbf{I} \end{bmatrix}$$

$$= \mathbf{I} - \eta_{\mathbf{x}} \begin{bmatrix} \mathbf{H}_{\mathbf{xx}} - \nabla_{\mathbf{x}}(\mathbf{G}_{\mathbf{xy}}\mathbf{G}_{\mathbf{yy}}^{-1}\nabla_{\mathbf{y}}f) & \mathbf{H}_{\mathbf{xy}} - \nabla_{\mathbf{y}}(\mathbf{G}_{\mathbf{xy}}\mathbf{G}_{\mathbf{yy}}^{-1}\nabla_{\mathbf{y}}f) \\ c\mathbf{G}_{\mathbf{yx}} & c\mathbf{G}_{\mathbf{yy}} \end{bmatrix} \begin{bmatrix} \mathbf{I} & \\ -\mathbf{G}_{\mathbf{yy}}^{-1}\mathbf{G}_{\mathbf{yx}} & \mathbf{I} \end{bmatrix}$$

$$= \mathbf{I} - \eta_{\mathbf{x}} \begin{bmatrix} \mathbf{H}_{\mathbf{xx}} - \mathbf{H}_{\mathbf{xy}}\mathbf{G}_{\mathbf{yy}}^{-1}\mathbf{G}_{\mathbf{yx}} - \nabla_{\mathbf{x}}(\square) + \nabla_{\mathbf{y}}(\square)\mathbf{G}_{\mathbf{yy}}^{-1}\mathbf{G}_{\mathbf{yx}} & \mathbf{H}_{\mathbf{xy}} - \nabla_{\mathbf{y}}(\square) \\ 0 & c\mathbf{G}_{\mathbf{yy}} \end{bmatrix},$$

where $\square$ is a shorthand for $\mathbf{G}_{\mathbf{xy}}\mathbf{G}_{\mathbf{yy}}^{-1}\nabla_{\mathbf{y}}f$. Let

$$\widetilde{\mathbf{H}}_{\mathbf{xx}} := \mathbf{H}_{\mathbf{xx}} - \mathbf{H}_{\mathbf{xy}}\mathbf{G}_{\mathbf{yy}}^{-1}\mathbf{G}_{\mathbf{yx}} - \nabla_{\mathbf{x}}(\square) + \nabla_{\mathbf{y}}(\square)\mathbf{G}_{\mathbf{yy}}^{-1}\mathbf{G}_{\mathbf{yx}}.$$

We can now see that the eigenvalues of $\mathbf{J}$ are exactly those of $\mathbf{I} - \eta_{\mathbf{x}}\widetilde{\mathbf{H}}_{\mathbf{xx}}$ and those of $\mathbf{I} - \eta_{\mathbf{y}}\mathbf{G}_{\mathbf{yy}}$. It follows that all eigenvalues of $\mathbf{J}$ are real.[6] Suppose that

$$\eta_{\mathbf{x}} < \frac{2}{\max\{\rho(\widetilde{\mathbf{H}}_{\mathbf{xx}}), c\rho(\mathbf{G}_{\mathbf{yy}})\}}.$$

In that case, if $(\mathbf{x}, \mathbf{y})$ is a local Stackelberg equilibrium, then from the second-order necessary condition, both $\widetilde{\mathbf{H}}_{\mathbf{xx}}$ and $\mathbf{G}_{\mathbf{yy}}$ are positive semidefinite. As a result, all eigenvalues of $\mathbf{J}$ would be in $(-1, 1]$. By Definition 2, this suggests that $(\mathbf{x}, \mathbf{y})$ is a stable fixed point.

On the other hand, if $(\mathbf{x}, \mathbf{y})$ is a strictly stable fixed point, then all eigenvalues of $\mathbf{J}$ fall in $(-1, 1)$, which suggests that $\widetilde{\mathbf{H}}_{\mathbf{xx}} \succ 0$ and $\mathbf{G}_{\mathbf{yy}} \succ 0$. By the sufficient condition, $(\mathbf{x}, \mathbf{y})$ is a local Stackelberg equilibrium. $\square$

**Remark 2.** *From the proof above, it is can be seen that "strict local Stackelberg equilibria", i.e. points that satisfy the sufficient conditions, must be strictly stable fixed points of (8). Then by Proposition 4, FR locally converges to such points. Thus, the set of points that FR locally converges to is the same as local Stackelberg equilibria, up to degenerate cases where the Jacobian spectral radius is exactly* 1.

## D EXPERIMENTAL DETAILS

### D.1 LOW DIMENSIONAL PROBLEMS

The algorithms we compared with are

$$\begin{bmatrix} \mathbf{x}_{t+1} \\ \mathbf{y}_{t+1} \end{bmatrix} \leftarrow \begin{bmatrix} \mathbf{x}_t \\ \mathbf{y}_t \end{bmatrix} - \eta \begin{bmatrix} \nabla_{\mathbf{x}}f(\mathbf{x}_t, \mathbf{y}_t) \\ -\nabla_{\mathbf{y}}f(\mathbf{x}_t, \mathbf{y}_t) \end{bmatrix}, \tag{GDA}$$

$$\begin{bmatrix} \mathbf{x}_{t+1} \\ \mathbf{y}_{t+1} \end{bmatrix} \leftarrow \begin{bmatrix} \mathbf{x}_t \\ \mathbf{y}_t \end{bmatrix} - 2\eta \begin{bmatrix} \nabla_{\mathbf{x}}f(\mathbf{x}_t, \mathbf{y}_t) \\ -\nabla_{\mathbf{y}}f(\mathbf{x}_t, \mathbf{y}_t) \end{bmatrix} + \eta \begin{bmatrix} \nabla_{\mathbf{x}}f(\mathbf{x}_{t-1}, \mathbf{y}_{t-1}) \\ -\nabla_{\mathbf{y}}f(\mathbf{x}_{t-1}, \mathbf{y}_{t-1}) \end{bmatrix}, \tag{OGDA}$$

$$\begin{bmatrix} \mathbf{x}_{t+1} \\ \mathbf{y}_{t+1} \end{bmatrix} \leftarrow \begin{bmatrix} \mathbf{x}_t \\ \mathbf{y}_t \end{bmatrix} - \eta \begin{bmatrix} \nabla_{\mathbf{x}}f(\mathbf{x}_t - \eta\nabla_{\mathbf{x}}f(\mathbf{x}_t, \mathbf{y}_t), \mathbf{y}_t + \eta\nabla_{\mathbf{y}}f(\mathbf{x}_t, \mathbf{y}_t)) \\ -\nabla_{\mathbf{y}}f(\mathbf{x}_t - \eta\nabla_{\mathbf{x}}f(\mathbf{x}_t, \mathbf{y}_t), \mathbf{y}_t + \eta\nabla_{\mathbf{y}}f(\mathbf{x}_t, \mathbf{y}_t)) \end{bmatrix}, \tag{EG}$$

$$\begin{bmatrix} \mathbf{x}_{t+1} \\ \mathbf{y}_{t+1} \end{bmatrix} \leftarrow \begin{bmatrix} \mathbf{x}_t \\ \mathbf{y}_t \end{bmatrix} - \eta \begin{bmatrix} \mathbf{I} & -\lambda\mathbf{H}_{\mathbf{xy}} \\ \lambda\mathbf{H}_{\mathbf{yx}} & \mathbf{I} \end{bmatrix} \begin{bmatrix} \nabla_{\mathbf{x}}f(\mathbf{x}_t, \mathbf{y}_t) \\ -\nabla_{\mathbf{y}}f(\mathbf{x}_t, \mathbf{y}_t) \end{bmatrix}, \tag{SGA}$$

$$\begin{bmatrix} \mathbf{x}_{t+1} \\ \mathbf{y}_{t+1} \end{bmatrix} \leftarrow \begin{bmatrix} \mathbf{x}_t \\ \mathbf{y}_t \end{bmatrix} - \eta \begin{bmatrix} \nabla_{\mathbf{x}}f(\mathbf{x}_t, \mathbf{y}_t) \\ -\nabla_{\mathbf{y}}f(\mathbf{x}_t, \mathbf{y}_t) \end{bmatrix} - \gamma\eta\nabla\|\nabla f(\mathbf{x}_t, \mathbf{y}_t)\|^2. \tag{CO}$$

We used a learning rate of $\eta = 0.05$ for all algorithms, $\lambda = 1.0$ for SGA and $\gamma = 0.1$ for CO. We did not find SGA with alignment (Balduzzi et al., 2018) to be qualitatively different from SGA in our experiments.

---

[6] $\widetilde{\mathbf{H}}_{\mathbf{xx}}$ is always symmetric.

## D.2 Mixture of Gaussian Experiment

**Dataset.** The mixture of Gaussian dataset is composed of 5,000 points sampled independently from the following distribution $p_{\mathcal{D}}(x) = \frac{1}{3}\mathcal{N}(-4, 0.01) + \frac{1}{3}\mathcal{N}(0, 0.01) + \frac{1}{3}\mathcal{N}(4, 0.01)$ where $\mathcal{N}(\mu, \sigma^2)$ is the probability density function of a 1D-Gaussian distribution with mean $\mu$ and variance $\sigma^2$. The latent variables $\mathbf{z} \in \mathbb{R}^{16}$ are sampled from a standard Normal distribution $\mathcal{N}(\mathbf{0}, \mathbf{I})$. Because we want to use full-batch methods, we sample 5,000 points that we re-use for each iteration during training. For the two-dimensional case, we generate the data from 9 Gaussians with $\mu_x \in \{-3, 0, 3\}$ and $\mu_y \in \{-3, 0, 3\}$. The covariance matrix is $0.01\mathbf{I}$.

**Neural Networks Architecture.** Both the generator and discriminator are 2 hidden layer neural networks with 64 hidden units and Tanh activations.

**Other Hyperparameters.** For FR, we use conjugate gradient (CG) in the inner-loop to approximately invert the Hessian. In practice, we use 10 CG iterations (5 iterations also works well). Since the loss surface is highly non-convex (let alone quadratic), we add damping term to stabilize the training. Specifically, we follow Levenberg-Marquardt style heuristic adopted in Martens (2010). For both generator and discriminator, we use learning rate 0.0002. For consensus optimization (CO), we tune the gradient penalty coefficient using grid search over $\{0.01, 0.03, 0.1, 0.3, 1.0, 3.0, 10.0\}$.

## D.3 MNIST Experiment

**Dataset.** The dataset we used in our experiment only includes class 0 and 1. For each class, we take 4,800 training examples. Overall, we have 9,800 examples. The latent variables $\mathbf{z} \in \mathbb{R}^{64}$ are sampled from a standard Normal distribution $\mathcal{N}(\mathbf{0}, \mathbf{I})$.

**Neural Networks Architecture.** Both the generator and discriminator are 2 hidden layer neural networks with 512 hidden units and Tanh activations. For each fully-connected layer, we use spectral normalization to stabilize training.

**Other Hyperparameters.** For FR, we use conjugate gradient (CG) in the inner-loop to approximately invert the Hessian. In practice, we use 5 CG iterations for computational consideration. We also use the same damping scheme as MOG experiment. For both generator and discriminator, we use learning rate 0.0001. We use batch size 2,000 in our experiments.

## D.4 Computing Correction Term on MOG and MNIST Experiments

The main innovation of this work is the introduction of the correction term which encourages both players (leader and follower) to stay close to the ridge. In the main paragraph, we focused on local convergence of our algorithm and essentially the Hessian matrix is constant. However, we know that the curvature (Hessian) of the loss surface might change rapidly in practice especially when both players are parameterized by deep neural networks, making the computation of Hessian inverse highly non-trivial. Here, we summarize detailed steps in computing $\eta_{\mathbf{x}}\mathbf{H}_{\mathbf{yy}}^{-1}\mathbf{H}_{\mathbf{yx}}\nabla_{\mathbf{x}}f$:

1. Computing $\eta_{\mathbf{x}}\mathbf{H}_{\mathbf{yx}}\nabla_{\mathbf{x}}f$ by finite difference $\mathbf{b} := \nabla_{\mathbf{y}}f(\mathbf{x}, \mathbf{y}) - \nabla_{\mathbf{y}}f(\mathbf{x} - \eta_{\mathbf{x}}\nabla_{\mathbf{x}}f, \mathbf{y})$;
2. Assigning $\mathbf{x} \leftarrow \mathbf{x} - \eta_{\mathbf{x}}\nabla_{\mathbf{x}}f$ such that the Hessian $\mathbf{H}$ below are evaluated at the updated $\mathbf{x}$;
3. Solving linear system $\left(\mathbf{H}_{\mathbf{yy}}^2 + \lambda\mathbf{I}\right)\Delta\mathbf{y} = \mathbf{H}_{\mathbf{yy}}\mathbf{b}$ using conjugate gradient to get an approximation of $\Delta\mathbf{y} = \left(\mathbf{H}_{\mathbf{yy}}^2 + \lambda\mathbf{I}\right)^{-1}\mathbf{H}_{\mathbf{yy}}\mathbf{b} \approx \eta_{\mathbf{x}}\mathbf{H}_{\mathbf{yy}}^{-1}\mathbf{H}_{\mathbf{yx}}\nabla_{\mathbf{x}}f$;
4. Adapting the damping coefficient $\lambda$ by computing reduction ratio
   $$\rho = \frac{\|\mathbf{b}\|_2^2 - \|\nabla_{\mathbf{y}}f(\mathbf{x}, \mathbf{y}) - \nabla_{\mathbf{y}}f(\mathbf{x} - \eta_{\mathbf{x}}\nabla_{\mathbf{x}}f, \mathbf{y} + \Delta\mathbf{y})\|_2^2}{\|\mathbf{b}\|_2^2 - \|\mathbf{H}_{\mathbf{yy}}\Delta\mathbf{y} - \mathbf{b}\|_2^2},$$
   which measures whether the loss surface is locally quadratic or not. We note that $\rho$ should be exactly 1 in the quadratic case if $\lambda = 0$. We then adjust the damping with Levenberg-Marquardt style heuristic (Martens, 2010) as follows:
   $$\lambda \longleftarrow \begin{cases} 1.1\lambda & \text{if } 0 < \rho \leq 0.5 \\ 0.9\lambda & \text{if } \rho > 0.95 \\ 2.0\lambda & \text{if } \rho \leq 0 \\ \lambda & \text{otherwise} \end{cases}$$

5. Setting $\Delta\mathbf{y} \leftarrow \mathbf{0}$ if $\rho \leq 0$ (essentially we don't believe the approximation if $\rho$ is negative).

When momentum or preconditioning is applied, we modify $\nabla_\mathbf{x} f$ above with the momentum or preconditioned version. To be specific, we apply momentum and preconditioning before the computation of the correction term, Here, we give an example of FR with momentum:

$$
\begin{aligned}
\begin{bmatrix} \mathbf{x}_{t+1} \\ \mathbf{y}_{t+1} \end{bmatrix} &\leftarrow \begin{bmatrix} \mathbf{x}_t \\ \mathbf{y}_t \end{bmatrix} - \begin{bmatrix} \mathbf{I} & \\ -\mathbf{H}_{\mathbf{yy}}^{-1}\mathbf{H}_{\mathbf{yx}} & \mathbf{I} \end{bmatrix} \begin{bmatrix} \eta_\mathbf{x}\nabla_\mathbf{x} f + \gamma\mathbf{m}_{\mathbf{x},t} \\ -\eta_\mathbf{y}\nabla_\mathbf{y} f + \gamma\mathbf{m}_{\mathbf{y},t} \end{bmatrix} \\
\begin{bmatrix} \mathbf{m}_{\mathbf{x},t+1} \\ \mathbf{m}_{\mathbf{y},t+1} \end{bmatrix} &\leftarrow \begin{bmatrix} \gamma\mathbf{m}_{\mathbf{x},t} + \eta_\mathbf{x}\nabla_\mathbf{x} f \\ \gamma\mathbf{m}_{\mathbf{y},t} - \eta_\mathbf{y}\nabla_\mathbf{y} f \end{bmatrix}
\end{aligned}
\tag{9}
$$

which is equivalent to Eqn.(3) in the quadratic case since it is a linear dynamical system. Nevertheless, we argue that it is more effective to use Eqn.(9) when the loss surface is highly non-quadratic.

## E  ADDITIONAL RESULTS

### E.1  THE ROLE OF PRECONDITIONING

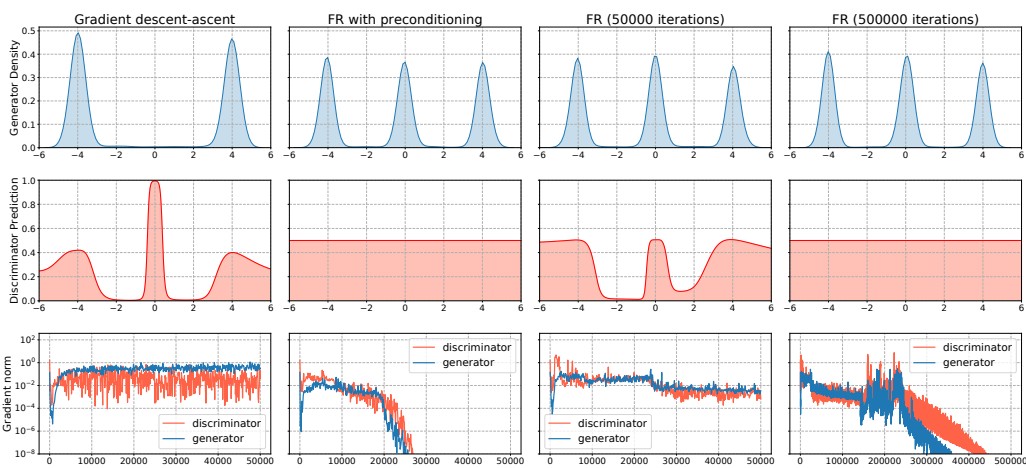

**Figure 9:** Ablation study on the effect of preconditioning. Vanilla FR also converges at the end of training though it takes much longer. The KDE plots use Gaussian kernel with bandwidth 0.1.

Following the same setting as Fig. 4, we investigate the effect of preconditioning for our algorithm. As we shown in section 4.1, FR is compatible with preconditioning with same theoretical convergence guarantee. In Fig. 4, we use diagonal preconditioning for accelerating the training. Here, we report the results of FR *without* preconditioning in Fig. 9. For fair comparison, we also tune the learning rate for vanilla FR and the optimal learning rate is 0.05. Our first observation is that vanilla FR does converge with 500,000 iterations which is consistent with our theoretical results. Particularly, the discriminator is being fooled at the end of training and the gradient vanishes. Our second observation is that it takes much longer to converge, which can be seen from the comparison between the second column (preconditioned version) and the third column. With the same time budget (50,000 iterations), preconditioned FR already converges as seen from the gradient norm curves while the vanilla FR is far from converged.

### E.2  THE ROLE OF MOMENTUM

In this subsection, we discuss the effect of momentum in our algorithm. We first consider the following quadratic problem:

$$
f(\mathbf{x}, \mathbf{y}) = -0.45x_1^2 - 0.5x_2^2 - 0.5y_1^2 - 0.05y_2^2 + x_1y_2 + x_2y_2.
$$

In this problem, $(\mathbf{0}, \mathbf{0})$ is a local (and global) minimax. We run FR with learning rate $\eta = 0.2$ and momentum values $\gamma \in \{0.0, 0.5, 0.8\}$, and observe how fast the iterates approach the origin.

We also compare FR with gradient descent-ascent in this problem. Note that when learning rate ratio (ratio of the follower's learning rate to the leader's learning rate) is 1, GDA diverges. We use

a grid search for the follower's learning rate $\eta_{\mathbf{y}} \in \{0.1, 0.2, 0.4, 0.8, 1.6\}$ and learning rate ratio $c \in \{5, 10, 20, 40, 80\}$. We experiment with momentum $\gamma \in \{0.0, \pm0.1, \pm0.2, \pm0.4, \pm0.8\}$. The best result for GDA without momentum is achieved by $\eta_{\mathbf{y}} = 0.8$, $c = 20$; the best result for GDA with momentum is achieved by $\eta_{\mathbf{y}} = 1.6$, $c = 40$ and $\gamma = 0.2$. The results are plotted in Fig. 10.

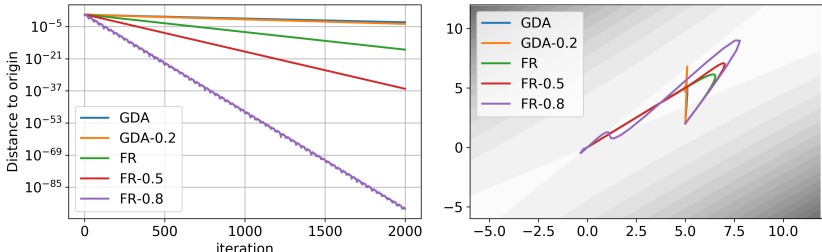

**Figure 10: Left:** distance to the origin for GDA, GDA with 0.2 momentum, FR, FR with 0.5 momentum and FR with 0.8 momentum; **Right:** trajectory of each algorithm; we plot the values of $x_1$ and $y_1$ and the contour for the function value on the plane $(x_1, 0, y_1, 0)$.

We can see that momentum speeds up FR dramatically. In contrast, GDA with momentum does not improve much over GDA without momentum. Moreover, under large momentum values (i.e. $\gamma = 0.8$), GDA diverges even when using very large learning rate ratios.

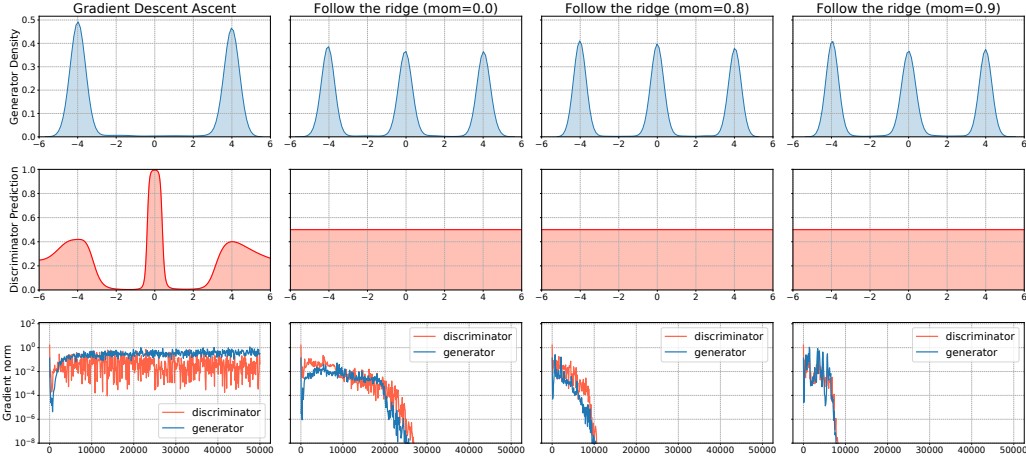

**Figure 11:** Empirical investigation on the effect of *positive* momentum. With larger momentum coefficient (e.g., $\gamma = 0.9$), the convergence of FR gets further improved. The KDE plots use Gaussian kernel with bandwidth 0.1.

We further test the acceleration effect of momentum on the mixture of Gaussian benchmark. Keeping all other hyperparameters the same as those used in Fig. 4, we conduct experiments with momentum coefficient $\gamma \in \{0.8, 0.9\}$. As shown in the gradient norm plots of Fig. 11, FR with large positive momentum coefficient converges faster than the one with zero momentum (the second column). Particularly, FR is able to converge within 10,000 iterations with $\gamma = 0.9$, yielding roughly a factor of 3 improvement in terms of convergence.

### E.3 SPECTRUM FOR GAN MODEL

As we claimed in Section 6.2.1, all eigenvalues of $\mathbf{H_{xx}} - \mathbf{H_{xy}H_{yy}^{-1}H_{yx}}$ are nonnegative while all eigenvalues of $\mathbf{H_{yy}}$ are non-positive. Here we plot all eigenvalues of them in log scale. To be noted, we plot the eigenvalues for $-\mathbf{H_{yy}}$ for convenience. As expected, the Hessian matrix for the

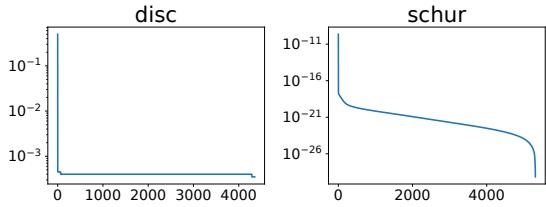

**Figure 12:** Top-20 eigenvalues.

discriminator is negative semi-definite while the Schur compliment is positive semi-definite.

