# OpenReview forum: "On Solving Minimax Optimization Locally: A Follow-the-Ridge Approach"
_ICLR.cc/2020/Conference — Accept (Poster)_

### Official Review · AnonReviewer2 · 2019-10-21
**Official Blind Review #2**

**Rating:** 6

**Review:**

Summary
The present work proposes a new algorithm, "Follow the Ridge" (FR) that uses second order gradient information to iteratively find local minimax points, or Stackelberg equilibria in two player continuous games. The authors show rigorously that the only stable fixed points of their algorithm are local minimax points and that their algorithm therefore converges locally exactly to those points. They show that the resulting optimizer is compatible with heuristics like RMSProp and Momentum. They further evaluate their algorithm on polynomial toy problems and simple GANs.

Decision
I think that this is a solid paper that addresses the well-defined goal of finding an optimizer that only converges to local minimax points. This is established based on both theoretical results and numerical experiments. Since there has been a recent interest in minimax points as a possible solution concept for GANs, I believe the paper should be accepted.

The paper occasionally makes claims that the solutions of GANs should consist of local minimax points ("We emphasize that GAN training is better viewed as a sequential game rather than the simultaneous game, since the primary goal is to learn a good generator."), which are not backed up by empirical results or reference to existing literature. If anything, the empirical results in this paper do not show improvement of the resulting generator (with the exception of the 1-dimensional example that has a particular rigidity since low discriminator output can easily restrict the movement of generator mass based on first order information). The right solution concept for GANs is not what the paper is about, but before publication the authors should remove these claims, identify them as speculative, or substantiate them .

Suggestions for revision
(1) In the last displayed formula on page 4 it should be the gradient w.r.t x.
(2) Remove, substantiate, or mark as speculative the claims regarding the right notion of solution concept for GANs.

Questions to the authors
(1) You write " There is also empirical evidence against viewing GANs as simultaneous games (Berard et al., 2019). ". Could you please elaborate, why Berard et al. provides empirical evidence against viewing GANs as simultaneous games?
(2) The Batch size for MNIST of 2000 is much larger than the values I have seen in other works. What is the effect of using more realistic batch sizes in training?
(3) When measuring the speed with which consensus optimization and FR converge, shouldn't you allow consensus optimization five times as many iterations, since you are using five iterations of CG to invert the Hessians in each step?
(4) You mention that you use CG to invert the Hessian, but the Hessian is not positive definite? Do you apply CG to the adjoint equations?

**Experience Assessment:**

I have published one or two papers in this area.

**Review Assessment: Checking Correctness Of Derivations And Theory:**

I assessed the sensibility of the derivations and theory.

**Review Assessment: Checking Correctness Of Experiments:**

I assessed the sensibility of the experiments.

**Review Assessment: Thoroughness In Paper Reading:**

I read the paper at least twice and used my best judgement in assessing the paper.

---

> ### Author Response · Authors · 2019-11-06
> **Response to Reviewer #2**
>
> Thank you for your detailed comments and kind words about our work.
>
> Regarding the claim that GANs training is a sequential game, we note that the majority of GAN papers consider GAN training as a divergence minimization problem. For example, f-GAN is minimizing f-divergence, the original GAN is minimizing JS-divergence and W-GAN is minimizing Wasserstein distance. By taking this perspective, we are implicitly modeling GAN training as a sequential game since the divergence interpretation involves solving the maximization in the inner loop. Hence minimax should be an appropriate solution concept. We will clarify this point in our next revision.
>
> A main observation of [1] is that when GANs are trained to generate good samples, the generator seems to be closer to a saddle point than a local minimum of the loss (see Figure 5 and 6). Thus the GANs are not at local Nash equilibria, but achieve good empirical performance. Since (local) Nash equilibrium is the typical solution concept for simultaneous games, we consider this an evidence against viewing GANs as simultaneous games.
>
> The reason why we used batch size 2,000 for MNIST is that our analysis was done for noiseless setting (full-batch). To exclude the factor of subsampling noise, we used large batch training. We leave the stochastic version of our algorithm to future work as it is highly non-trivial.
>
> Regarding the comparison with consensus optimization, we measured the training steps instead of wall-clock time. In terms of wall-clock time, consensus optimization does take fewer computation at each step. But we note that is not the main focus of our work.
>
> Regarding how we invert the Hessian, we discussed the details in section 6.2.1 in our submission and we've added more details in the Appendix D.4. Specifically, we solve the linear system $\mathbf{H}_{yy}^2 z = \mathbf{H}_{yy} \mathbf{H}_{yx} \nabla_{x} f$.
>
>
>
> [1] Berard et al., "A closer look at the optimization landscapes of generative adversarial networks", 2019.

---

> > ### Comment · AnonReviewer2 · 2019-11-14
> > **Thanks for the reply**
> >
> > Thanks for the reply. You write
> >
> > "A main observation of [1] is that when GANs are trained to generate good samples, the generator seems to be closer to a saddle point than a local minimum of the loss (see Figure 5 and 6). Thus the GANs are not at local Nash equilibria, but achieve good empirical performance. Since (local) Nash equilibrium is the typical solution concept for simultaneous games, we consider this an evidence against viewing GANs as simultaneous games."
> >
> > This is indeed one possible interpretation, but not arguably not the only one. I would therefore suggest giving the reader the above argument and letting them decide for themselves. The way it is written presently, it might be misunderstood as [1] investigating the question of simultaneous vs sequential game, which it does not.
> >
> > On page 1, you also write
> > "Unlike simultaneous games, many practical machine learning algorithms, including generative adversarial networks
> > (GANs) (Goodfellow et al., 2014; Arjovsky et al., 2017),
> > adversarial training (Madry et al., 2018) and primal-dual
> > reinforcement learning (Du et al., 2017; Dai et al., 2018),
> > explicitly specify the order of moves between players and
> > the order of which player acts first is crucial for the problem."
> >
> > This, again, is a bit misleading since in the case of GANs, for instance, simultaneous gradient descent is commonly used and was recommended, for instance, in the tutorial of goodfellow (https://arxiv.org/pdf/1701.00160.pdf page 20).
> >
> > I'm sorry if I'm appearing pedantic, but the question of the right notion of game underlying GANs is an interesting research question that deserves its own treatment.
> > To my knowledge there is no firm evidence one way or the other and therefore preliminary claims should be avoided.
> > I would appreciate if the authors could modify passages in question (or tell me why they disagree)

---

> > > ### Author Response · Authors · 2019-11-15
> > > **Further Response**
> > >
> > > Thank you for all the comments and suggestions.
> > >
> > > In terms of our debate around the right notion of game underlying GANs, we agree that this issue is controversial, so we will modify our paper as it's not the main focus of our paper. We have already toned down our claim in the related work section in the current revision. We were spending most of our time discussing with reviewer #1 and didn't get enough time to go through the paper to modify all those statements, and are really really sorry about that. We promise that we will update the paper according to your suggestions in the camera ready version should our paper be accepted.

---

### Official Review · AnonReviewer1 · 2019-10-22
**Official Blind Review #1**

**Rating:** 6

**Review:**

Summary: This paper designs a set of dynamics for learning in games called follow-the-ridge with the goal of finding local stackelberg equilibria. The main theoretical results show that the only stable attractors of the dynamics are stackelberg equilibria. Moreover, the authors give a deterministic convergence rate for the vanilla algorithm and a convergence rate using momentum. Empirical results show the learning dynamics cancel out rotational components and drive the vector field to zero rapidly, while reaching good performance on simple GAN examples.

Review: This paper focus on sequential games, which is the common formulation of GANs and a number of games in machine learning applications. From this perspective, it is natural to look at Stackelberg equilibria. In my opinion, the objective of the paper is important and relevant. The theoretical and empirical results are reasonably convincing. However, I do have some rather serious concerns about the general-sum game results and several questions regarding the relation to related work and the experiment details that need to be addressed.

1. The FR dynamics in algorithm 1 are closely related to the dynamics in [1]. In particular, the Jacobian of the FR dynamics is a similarity transform of the Jacobian of the dynamics in [1]. As a result, each algorithm has the same set of stable attractors. This should probably be mentioned in the paper. Given this relation, it is not clear what the advantage of the FR dynamics are over the dynamics in [1]. Could you please discuss this?

2. The gradient penalty regularization connection does not make sense in section 4.1. The optimization problem presented has an issue because the dimensions do not align in the constraint. The quantity \nabla_x f(x, y)^T H_yy^{-1}\nabla_x f(x, y) would not be defined if the dimensions of the players are not equal.

3. In the related work it is claimed that two time-scale GDA converges only to local minimax and [2] is cited. I would avoid using this claim with respect to that paper since the statement following the main result in the paper is not right (see proposition 11 of [3] for proof). It is not clear what is meant when it is claimed that [1] can converge to non-local Stackelberg points. The dynamics in [1] only converge to local minimax points in the special case of zero-sum games.

4. Since the dynamics in the paper are the closest to those in [1], it seems that the paper would be stronger by comparing with that set of dynamics.

5. I found it to be quite impressive that the vector field is driven to zero in the GAN examples. Just to clear, for each algorithm when the ‘gradient norm’ is shown, does this mean the norm of the update for each norm or does it mean the individual derivative for each player. For example in FR, would it be the norm of the derivative with respect to the follower variable of the function or the norm of the update including the second order information?

6. The path angle plot was interesting to see for the GAN example. The authors claim that the eigenvalues of the second order equilibria condition are non-negative. It would be nice if the authors could show the eigenvalues in the appendix and discuss how they were computed since it may be non-trivial to compute depending on the network size.

7. The damping method to stabilize training is not quite clear. Could you provide more details about how this was done?


My primary concerns have to do with the portion of the paper considering general-sum games. I do not understand where proposition 7 and 8 come from. I am not convinced the definitions provided are necessary and sufficient conditions for Stackelberg equilibria. In [1], a differential Stackelberg equilibrium is defined. The definition in this paper does not appear to agree with the definition in [1]. The final positive definite condition in proposition 7 and 8 does not appear to be taking the total derivative 2 times when I evaluate the derivatives, so I am not sure what the quantity is. If this is not a proper set of conditions for the equilibria, then it would also mean that the dynamics do not only converge to equilibria in general-sum games. It is important that the authors clear up this concern since I do not believe Theorem 3 holds as a consequence of problems with propositions 7 and 8.

[1] Fiez et al.,  "Convergence of Learning Dynamics in Stackelberg Games",  2019.
[2] Heusel et al., "GANs trained by a two time-scale update rule converge to a local Nash equilibrium", 2017.
[3] Mazumdar et al.,  "On Finding Local Nash Equilibria (and only Local Nash Equilibria) in Zero-Sum Games", 2019.

Post Author Response: Thanks to the authors for the effort in discussing the paper with me. The authors made several changes to the paper in response to my comments including removing section 4.1, fixing comments about related work, including details on the damping procedure, showing experimental comparisons to [1] along with an explanation of why the dynamics in this paper may be preferred for training GANs, providing details on propositions 7 and 8 and including reference to [1], adding further assumptions on the functions, and attempting to make theorem 3 more clear. Overall, I think this paper proposes an interesting set of dynamics, several meaningful theoretical guarantees, and impressive empirical results. I would be curious to see how it performs on even more large-scale GAN problems in the future. As a result, I have changed my original score from a weak reject to a weak accept. My primary concerns with the paper regarded the general-sum convergence results and I appreciated the explanations from the authors. I am still of the opinion that theorem 3 could be stated more rigorously in the sense that the neighborhood on which the local convergence holds should be more explicit. It seems to me that the convergence result may only hold in a ball around an equilibrium in which the implicit function is well-defined and the FR dynamics will be attracted to r(x) and that this space could be arbitrarily small for some problems. Nonetheless, this result is only in the appendix, and the paper includes enough contributions beyond this to warrant acceptance.


**Experience Assessment:**

I have published one or two papers in this area.

**Review Assessment: Checking Correctness Of Derivations And Theory:**

I carefully checked the derivations and theory.

**Review Assessment: Checking Correctness Of Experiments:**

I carefully checked the experiments.

**Review Assessment: Thoroughness In Paper Reading:**

I read the paper thoroughly.

---

> ### Author Response · Authors · 2019-11-06
> **Response to Reviewer #1**
>
> Thank you for your detailed comments and feedback. We hope that our responses below adequately resolve your concerns. Although we believe the current revision does a much better job of presenting these arguments, we warmly encourage you to provide any criticisms that may help us further express these points more clearly.
>
> - Regarding the comparison with [1]:
>
> We are aware of and has cited the work in [1], and agree that more discussion is due. Compared to the dynamics in [1] (that is, Eqn (1) without noise), we believe that FR has two advantages. First, FR has a more intuitive interpretation, as it is trying to follow the follower's best response function, i.e., the ridge. In comparison, when positioned on the ridge, the update of the dynamics in [1] coincides with gradient descent-ascent and will drift away from the ridge. Second, the guarantees of FR carries over to non-zero-sum games. We believe this is important since it is a much more general setting and has applications such as hyperparameter tuning. In comparison, the authors of [1] acknowledged that in non-zero-sum games, their algorithm can converge to a point that is not differential/local Stackelberg equilibrium (see Remark 3 [1]).
>
> We also note that the main focus of [1] is different, which is proving convergence of gradient dynamics in the presence of noise. However, to do so, they assume that there is only one local maximum for the follower (Assumption 2), and that there is a timescale separation between the leader and the follower. Because of these two assumptions, their main results are not directly applicable to many problems including GAN training. Roughly speaking, they showed stronger results under stronger assumptions.
>
>
> - Regarding the definition of local Stackelberg equilibrium:
>
> Our definition for local Stackelberg equilibrium agrees with the definition in [1] up to edge cases. In particular, the Hessian of $\phi(x) := f(x, r(x))$ is exactly $D^2f_1$ in Definition 4, [1]. Thus, requiring $D^2f_1$ to be positive semi-definite is almost the same as requiring $x$ to be a local minimum of $\phi(x)$.
>
> We apologize for stating Proposition 7 and 8 without further explanation. We have updated our paper with a detailed explanation for their derivation.
>
> In particular, we can see that $\nabla\phi(x)=\nabla_{x}f+\nabla r(x)^T\nabla_{y}f$, which is the same as the transpose of $Df(x,r(x))$ using the notation in [1]. It then follows that $\nabla^2\phi(x)=\nabla_{x}(\nabla_{x}f+\nabla r(x)^T\nabla_{y}f)+\nabla_{y}(\nabla_{x}f+\nabla_{x}r(x)^T\nabla_{y}f)\nabla r(x)=DD f$. Substituting $\nabla r(x)$ with $-G_{yy}^{-1}G_{yx}$ would prove Proposition 7 and 8.
>
> - Regarding the gradient penalty regularization, we removed this section as it has little connection with our main contributions.
>
> - Regarding the gradient norm, we meant the individual derivative for each player.
>
> - Regarding the detailed damping scheme, we added the details in Appendix D.4 of current revision.
>
> - Regarding the eigenvalues of the second order equilibria condition, we are able to compute the Hessian and its inverse exactly for networks we used for mixture of Gaussian experiments. The networks we used were 2-hidden-layer MLP with 64 hidden units for each layer. To be specific, we compute each row of the Hessian by multiplying the Hessian with a vector (with all entries $1$). To be noted, the Hessian-vector product can be done efficiently by doing reverse-mode autodiff (backpropgation) twice. For exact values, we notice that the eigenvalues of the generator's Hessian are all zero (which is not surprising since the discriminator outputs a flat line) while the eigenvalues of the discriminator's Hessian are all positive (as we added $L_2$ regularization). Therefore, it is easy to see that the Schur compliment $H_{xx} - H_{xy}H_{yy}^{-1}H_{yx} = - H_{xy}H_{yy}^{-1}H_{yx}$ is positive definite.
>
> - Regarding the local minimax convergence claims in other works: It is indeed our mistake to cite [2] for our claim; we have removed this citation in our revision. However, we do not believe Proposition 11 in [3] contradicts our claim. In their example, the min player moves faster than the max player, so the dynamics converge to a local maximin. By "[1] can converge to non-local Stackelberg points", we meant that stable limit points of [1] can be points that are not local Stackelberg, which is acknowledged by the authors of [1] in Remark 3.
>
>
>
> [1] Fiez et al.,  "Convergence of Learning Dynamics in Stackelberg Games",  2019.
> [2] Heusel et al., "GANs trained by a two time-scale update rule converge to a local Nash equilibrium", 2017.
> [3] Mazumdar et al.,  "On Finding Local Nash Equilibria (and only Local Nash Equilibria) in Zero-Sum Games", 2019.

---

> > ### Comment · AnonReviewer1 · 2019-11-13
> > **Response**
> >
> > Sorry for the delayed response.
> >
> > Comments on the general-sum results:
> >
> > I appreciate the explanation on proposition 7 and 8. I do however, still have some significant concerns. To begin, I do not think you should be stating the propositions as a result in this paper given that the conditions are already given in [1]. The primary concern involves the analysis of the Jacobian for the FR dynamics in general-sum games. It seems that the results you have only hold if the follower is playing an exact best response along each step of the learning trajectory, that is y = r(x) for every x iterate. This would correspond to a set of decoupled dynamics where the only solution being learned is for the leader. Even though the dynamics push the follower to be close to r(x) it is not always exactly at r(x). To fix this issue and show the dynamics using y coincide with using r(x) you would need to use singular perturbation theory and introduce a timescale separation between the players. For this reason, I do not believe that Theorem 3 holds. If you do not agree with this line of reasoning, please feel free to explain why you believe it holds and hopefully we can discuss and resolve things before the discussion period ends. I am also happy to elaborate on my statements overhead if it is not clear what the issue is that I am raising.
> >
> > - I have to respectfully disagree that the dynamics in this paper are more intuitive than the dynamics in [1]. The dynamics from that paper directly reflect the underlying game structure. However, I do see some utility in the interpretation in this paper. Since the dynamics have the same critical points in [1], similar deterministic guarantees for that set of dynamics could be given. For this reason, I am not sure that giving the deterministic convergence results is any more significant that stochastic results. The results in [1] also gives guarantees under relaxed assumptions beyond what you mentioned. I do not agree with the claim that timescale separation makes results not applicable to GANs since to my knowledge they are often trained with a timescale separation. This essentially what makes me skeptical about the level of contribution in zero-sum games.
> >
> > - Thanks for fixing the issue with the gradient penalty regularization by removing it in the paper.
> >
> > - Why do you plot the gradient norm for each player instead of the norm of the vector field? The norm of the individual derivative does not indicate if it is at a critical point or not.
> >
> > - Thanks for including the details regarding the damping scheme in the paper and explaining how the eigenvalues were computed. I noticed that you have not included a plot of the eigenvalues in the paper yet. I believe that would be good to do. That is interesting to know that the generator hessian eigenvalues are all zero since that would correspond to the realizable assumption made in some GAN papers.
> >
> > - That is a good point that the example in [3] does not directly contradict your claim since all stable critical points of simultaneous gradient descent in zero sum games will be stackelberg. That being said, I appreciate you removing your statement since it is not known that all non-nash stable critical points of the simultaneous gradient dynamics are stackelberg equilibrium in zero-sum games in higher dimensions.
> >
> > - While you may say that [1] can converge to non-stackelberg points, the statement in remark 3 only regards general-sum games. The way it is stated in the paper now, it seems that you are saying the dynamics in [1] can converge to non-stackelberg points in zero-sum games. I believe you could edit this section to make it clear in zero-sum games, the dynamics in [1] will only reach critical points which are stackelberg, just as in this paper.
> >
> > - Finally, in point 4 from my review, I was trying to ask why you did not compare empirically to the dynamics in [1] since they are the closest to the dynamics in this paper and have similar properties. Sorry that this was not clear.

---

> > > ### Author Response · Authors · 2019-11-14
> > > **Further Response [2/2]**
> > >
> > > - About the contribution of this paper in zero-sum-games: On a second read we now agree that the some stochastic results in [1], namely remark 4, hold under less restrictive assumptions. However, we do not believe this undermines our contribution significantly, even in the zero-sum setting.
> > >
> > > Among other things, on the theory side, we: 1. proposed a novel algorithm and shown local convergence result for it (Theorem 1); 2. incorporated preconditioning and momentum into the algorithm with the same convergence guarantees (Proposition 6 and Theorem 2); 3. gave explicit local convergence rates (Theorem 2). None of these are corollaries of results in [1], and they hold under arbitrary constant learning rate ratios, which is much more favorable in practice.
> > >
> > > On the practical side, we demonstrated effectiveness of our algorithms in both simple examples and GANs. In the GAN setting in particular, we show that the gradient norm quickly diminishes to zero. In contrast, the algorithm in [1] does not achieve this in GAN training (see details below).
> > >
> > > For these reasons, we believe the contribution of our work is significant. This is not to mention our result for non-zero-sum games (Theorem 3).
> > >
> > > - About empirical comparison of FR and algorithm in [1]: We provide empirical results in GANs comparing FR and the dynamics in [1]. Here is the link for the results (https://docs.google.com/document/d/1HQbbd3nphY9bmhAu8Mir1M5gqpo5ZnlmewrVNVNdbsM/edit?usp=sharing). Currently, we find that the algorithm in [1] doesn't perform as well as FR (cf. Fig. 4 fourth column). In particular, the discriminator's output is not constant, and gradient norm does not decrease to zero. To be noted, we cannot exclude the factor of hyperparameter tuning for now, but we have tried our best to tune their method before the rebuttal deadline.
> > >
> > > - About the timescale separation, the majority of modern GAN papers use roughly same scale learning rates for both the generator and discriminator (e.g., BigGAN [2]). In [1], the learning rate of the generator has to be infinitely smaller than that of the discriminator. In practice, it potentially leads to mode collapse issue when a much smaller learning rate is used for the generator since the discriminator becomes too good and the gradient w.r.t the generator vanishes (see WGAN [3] paper for more discussions).
> > >
> > > - About plotting the gradient norm: First, we would like to point out that points where individual gradients are 0 have to be fixed points of FR. Note that when gradients are 0, all updates of FR are zero since the preconditioners are positive definite. In fact, the set of fixed points of FR is exactly the points where individual gradients are 0 (which is argued in our proof of Theorem 1). We are confused about the reviewer's remark that "The norm of the individual derivative does not indicate if it is at a critical point or not".
> > >
> > > Second, we mean to use the gradient norm of different algorithms as a measure for convergence (see e.g. Fig. 5). These algorithms use different update rules, but are all based on individual gradients. Therefore, we believe that it makes more sense to plot the norm of individual gradients for a comparison.
> > >
> > > - Regarding the description of the results in [1]: We would clarify this in our next revision.
> > >
> > > - Regarding the eigenvalues, we plot the top-20 eigenvalues (sort according to the magnitude) for the Hessian of the discriminator and the Schur compliment in Appendix E.3.
> > >
> > >
> > > Lastly, we thank reviewer #1 for all the comments again. We hope our responses address your concerns, especially your primary concern on Theorem 3.
> > >
> > >
> > >
> > > [1] Fiez et al.,  "Convergence of Learning Dynamics in Stackelberg Games", https://arxiv.org/pdf/1906.01217v2.pdf
> > > [2] Brock et al., "Large Scale GAN Training for High Fidelity Natural Image Synthesis", https://arxiv.org/pdf/1809.11096.pdf
> > > [3] Arjovsky et al., "Wasserstein GAN", https://arxiv.org/pdf/1701.07875.pdf

---

> > > > ### Comment · AnonReviewer1 · 2019-11-14
> > > > **Response Regarding Non-General-Sum Things**
> > > >
> > > > Before responding, I want to point out that [1] is the closest paper in the literature to this paper and so naturally it would be what the majority of the comments regarding related work would concern. I also want to make clear that my primary concerns in the review process have concerned the general-sum results, which is an independent issue. Moreover, I was not asking for experimental comparisons to the dynamics in [1] during the rebuttal period since this would be time consuming for you in a short time window and unreasonable to request. Sorry if that was not clear. I was simply making the point that the dynamics are similar for zero-sum games and so it may have been the most natural experimental comparison in the initial submission that could have made the paper stronger. With that being said, I do believe the example you provided along with the reasoning for the empirical advantage of FR makes sense and provides justification for why the dynamics may be preferable for training GANs.
> > > >
> > > > - You are right about the gradient norm that it should reflect the same thing, thank you for pointing this out to me. As I stated in my initial review, the fact that the FR dynamics quickly drive the gradient norm to zero is impressive in my opinion. I am, and have been, of the opinion that the empirical results are strong and the example you provided along with the reasoning you provided reinforces this fact.
> > > >
> > > > - Thanks for including the figure with the eigenvalues. However, I am not sure why you showed the top eigenvalues instead of the top and bottom eigenvalues since that is what would allow for evaluation of definiteness. It is not super clear to me that the Schur complement is actually positive from what you have shown. If I am missing something in this line of reasoning, please let me know.
> > > >
> > > > - Thanks for reiterating your convergence results in zero-sum games. I did not mean to sound dismissive of them in my previous comment. My intention was to try and understand why the FR dynamics would be preferred to the Stackelberg gradient dynamics in zero-sum games. The example you provided and the justification gives some evidence for why it may be the case. You are right that that the deterministic results are not direct corollaries of any results in [1], you provide interesting results using momentum and preconditioning, and you devised a set of dynamics that inherit the properties from the Stackelberg gradient dynamics but may come with empirical advantages. I acknowledge that this set of contributions is meaningful.
> > > >
> > > > I will now work on explaining further the potential issue with the general-sum results I see and post shortly.

---

> > > > > ### Author Response · Authors · 2019-11-14
> > > > > **Thank you for quick reply!**
> > > > >
> > > > > First, we really appreciate your further comments and clarifications. Sorry if our tone was a bit aggressive.
> > > > >
> > > > > Regarding the figure with the eigenvalues, we will update the paper once you post the response for Theorem 3. In the next version, we will include all eigenvalues. I meant to say positive semi-definite and sorry for the confusion.
> > > > >
> > > > > We're looking forward to discussing more with you on Theorem 3.

---

> > > ### Author Response · Authors · 2019-11-14
> > > **Further Response [1/2]**
> > >
> > > Thank you for the detailed feedback! We will try to answer your questions more clearly this time.
> > >
> > > - About Proposition 7 and 8: We believe these results are straightforward results of the definition of local Stackelberg equilibrium, which we already acknowledged to be proposed in [1] (see page 5). We've added another reference to [1] before Definition 4 for further clarity.
> > >
> > > - About the correctness of Theorem 3: The dynamics we consider is always (8), where all derivatives are evaluated at $(x_t,y_t)$, so we do not require the follower to play the best response in the algorithm. We would like to emphasize that $r(x)$ is the global solution to a non-convex problem that we never explicitly use. We would like to breakdown the proof of Theorem 3 so that the reviewer may point out which step does not hold if under the dynamics of (8).
> > >
> > > Step 1: We care about the fixed points of (8), and since the preconditioner is invertible, a fixed point of (8) necessarily satisfies $\nabla_{y}g(x,y)=0$ and $D_x f(x,y)=0$.
> > >
> > > Step 2: At a point such that $\nabla_{y}g(x,y)=0$ and $D_x f(x,y)=0$, the Jacobian of (8) can be calculated to be the last equation on page 16.
> > >
> > > Step 3: The eigenvalues of the Jacobian of (8) at such a point can be shown to be those of $I-\eta_{x}\tilde{H}_{xx}$ and $I-\eta_{y}G_{yy}$.
> > >
> > > We cannot see which step won't hold under our dynamics, namely Eqn (8). We kindly welcome the reviewer to point out which step is incorrect under the current dynamics. Otherwise, we should conclude that Theorem 3 is mathematically correct. If you are still concerned about Theorem 3, we are happy to discuss more before the rebuttal deadline.
> > >
> > >
> > > [1] Fiez et al.,  "Convergence of Learning Dynamics in Stackelberg Games", https://arxiv.org/pdf/1906.01217v2.pdf

---

> > > > ### Comment · AnonReviewer1 · 2019-11-15
> > > > **Response Regarding General-Sum Result**
> > > >
> > > > Thanks for the response on the general-sum results and going through your thoughts with me. I will try to be more clear and concrete in this explanation. Again, please let me know if something is unclear following this response or if you believe there is an issue with my reasoning.
> > > >
> > > > To begin, I agree with you that there is nothing mathematically wrong in the steps of your proof. However, I believe that there is a problem with the conclusions you draw from the steps in your proof regarding the convergence guarantees of the FR dynamics. Let $x$ be the player 1 variable and $y$ be the player 2 variable and $f$ be the objective of player 1 and $g$ be the objective of player 2. To summarize my argument, consider $(x_1^{\ast}, y_1^{\ast})$ to be a critical point of the FR dynamics and $(x_2^{\ast}, r(x_2^{\ast}))$ to be a stackelberg equilibrium where $x_2^{\ast} = x_1^{\ast}$, it can be that $y_1^{\ast} \neq r(x_2^{\ast})$ which would contradict the statement of Theorem 3. To motivate the more detailed analysis consider the following example. Suppose that $\nabla_{yy}^2g(x, r(x))$ has positive eigenvalues everywhere, but anywhere off of it you have negative eigenvalues. Let us further consider that the change from positive eigenvalues to negative eigenvalues is continuous so that for any small neighborhood a perturbation of $r(x)$ drives you away from the equilibrium. Then the only initial conditions for which the FR dynamics would reach the equilibrium would be given an initialization of $r(x)$, which is a set of measure zero. Now I go into the details of why I believe that your result may only hold if you explicitly assume the FR dynamics use $r(x)$ or if you introduced a timescale separation carefully.
> > > >
> > > > Consider what the implicit function theorem (IFT), which says the following (where for only this statement $f$ is an arbitrary function and not the cost function of player 1):
> > > >
> > > > Let $U \subset E$, $V \subset F$ be open and $f : U \times V \rightarrow G$ be $C^q$, $q \geq 1$. For some $x_0\in U$, $y_0 \in  V$ assume the partial derivative in the second argument $\nabla_{y}f(x_0,y_0) : F \rightarrow G$ is an isomorphism. Then there are neighborhoods $U_0$ of $x_0$ and $W_0$ of $f (x_0 , y_0 )$ and a $r:U_0\times W_0 \rightarrow V$ such that for all $(x,w)\in U_0\times W_0$, $f(x,r(x,w))=w$.
> > > >
> > > > Now consider a follower update
> > > > $$y_{t+1}=y_t-\gamma_2 \nabla_y g(x_t,y_t)$$
> > > > and suppose at a critical point $(x_0,y_0)$
> > > > $$\nabla_y g(x_0,y_0)=0$$
> > > > and $\nabla_{yy}^2g(x_0,y_0)$ is non-degenerate (hence, an isomorphism). Then IFT says that there exists
> > > > $$\exists \ U_0, \ W_0, \ r \in C^q(U_0\times W_0, V)$$
> > > > such that $x_0\in U_0$, $\nabla_yg(x_0,y_0)\in W_0$ and such that
> > > > $$\nabla_yg(x,r(x,0))=0, \ \forall x\in U_0$$
> > > > However, even on $U_0$, this does not mean that the $y$ iterates generated by the system
> > > > $$x_{t+1}=x_t-\gamma_1h_1(x_t, y_t))$$
> > > > $$y_{t+1}=y_t-\gamma_2 \nabla_yg(x_t,y_t)$$
> > > > coincide with the $r(x)$ iterates when player 1 is doing
> > > > $$x_{t+1}=x_t-\gamma_1h_1(x_t,r(x_t)).$$
> > > > where for simplicity we can just consider $h_1$ to be the total derivative of $f$.
> > > >
> > > > This same line of reasoning should analogously apply to the FR dynamics. So this is effectively why I believe that your results would only hold if the dynamics were explicitly using $r(x)$. To fix this issue, I think you would need to show that $r(x)$ is attracting on the ball you get from the IFT and this would also require the timescale separation. Essentially, the issue in the conclusion given in Theorem 3 is why singular perturbation theory is needed. A good reference that may make things more clear is [4].
> > > >
> > > > So to summarize, I don't think what has been stated in Theorem 3 is what has been proved or it has not been stated and the underlying assumptions given rigorously enough.
> > > >
> > > > Thanks for being willing to hear out my concerns regarding this result and having a cordial dialogue about it. Please let me know if you do not agree with my statements or if you have any questions.
> > > >
> > > > [4] Skinner, "Singular Perturbation Theory"

---

> > > > > ### Author Response · Authors · 2019-11-15
> > > > > **Elaborate a bit about the notations.**
> > > > >
> > > > > Thanks you for the explanation. Nevertheless, we feel rather confused about your notations in IFT part.
> > > > >
> > > > > - What do $x$ and $y$ mean? The leader and the follower?
> > > > >
> > > > > - What's the definition of $f$ and $g$? Is $f$ the objective of the leader while $g$ the objective of the follower?
> > > > >
> > > > > - Why you sometimes use $h$, sometimes $h_1$ and $h_2$? What's the difference between them?
> > > > >
> > > > > - What's the definition of $r$? Is it the same as ours? Why it sometimes take two arguments and sometimes just one?
> > > > >
> > > > > - What does $D_2$ mean?
> > > > >
> > > > > - Why would you expect the update of $y_t$ (the second last equation) coincides with $x_t$ (the last equation) if they are different players?

---

> > > > > > ### Comment · AnonReviewer1 · 2019-11-15
> > > > > > **Clarifications**
> > > > > >
> > > > > > Is it more clear now?

---

> > > > > > > ### Author Response · Authors · 2019-11-15
> > > > > > > **Yes**
> > > > > > >
> > > > > > > We're working on our response now, will get back to you very soon.

---

> > > > > ### Author Response · Authors · 2019-11-15
> > > > > **Your reasoning about best-response gradient dynamics does NOT apply to FR dynamics. Please pinpoint which step of our proof is wrong or give a specific example where FR fails.**
> > > > >
> > > > > Thank you for your presenting your arguments. Nevertheless, we are still not convinced of the need to use timescale separation. We also believe that there is no discrepancy between our statement of Theorem 3 and our claim in the main text.
> > > > >
> > > > > - An minor unstated assumption:
> > > > >
> > > > > We have noticed that we indeed made a mistake in our results in that, to apply Proposition 4, one needs the update rule to be differentiable. This in turn requires the objective function(s) to be thrice differentiable (in Assumption 1 we stated twice differentiable). We will modify Assumption 1 in our next revision.
> > > > >
> > > > > - About the example you mentioned in the second paragraph:
> > > > >
> > > > > The example you bring up is very interesting. However, we believe that as long as the update rule of FR is differentiable so that its Jacobian changes smoothly, we will be able to apply Proposition 4, i.e. use Taylor expansion at a equilibrium $(x^*,y^*)$. In that case, how Hessian changes becomes immaterial, as that does not affect the Jacobian of the update rule. In other words, our results will hold in your example if the cost functions are thrice differentiable.
> > > > >
> > > > > - Connecting the result of Theorem 3 to our local convergence claim:
> > > > >
> > > > > Our claim for local convergence is basically: there exists a neighborhood $U$ with $(x^*,y^*)\in U$ such that when initialized in $U$, FR converges to $(x^*,y^*)$. We believe such convergence follows from Theorem 3 just as the non-zero-sum convergence claim follows from Theorem 1. In particular, from Theorem 3, we know that at Stackelberg equilibria, the Jacobian eigenvalues fall in $[-1,1]$. Suppose that we consider a "strict Stackelberg equilibrium" $(x^*,y^*)$, i.e. $\tilde{H}_{xx}$ and $G_{yy}$ are both positive definite, then the Jacobian eigenvalues would fall in $(-1,1)$. Then, from Proposition 4 (well known result), there exists a neighborhood $U$ of $(x^*,y^*)$ such that when initialized in $U$, the following iterates will be contractions in $\Vert\cdot\Vert_2$.
> > > > >
> > > > > We kindly welcome the reviewer to point out the discrepancy between Theorem 3 and our claim of local convergence in the line of reasoning above.
> > > > >
> > > > > Note: "Strict Stackelberg equilibrium" is essentially the definition for differentiable Stackelberg equilibrium in [1].
> > > > >
> > > > > - About the arguments you presented about IFT:
> > > > >
> > > > > The reasoning you give concerns best-response gradient dynamics. However, FR is fundamentally different from best-response gradient dynamics, so currently we do not see how the analogy works for FR.
> > > > >
> > > > > First, what FR does in general-sum games is applying a correction to best-response gradient dynamics. Recall that GDA (with best-response gradient) is problematic, but FR's correction of GDA leads to satisfactory convergence. Therefore, problems with best-response gradient dynamics (the example you mentioned) does not apply to FR at all. We hope the reviewer to read our previous response about the relationship between FR and best-response gradient dynamics. They're fundamentally different.
> > > > >
> > > > > Second, the problem you raised is exactly one of the motivations underlying FR (see Section 4, although it is stated for zero-sum games). The correction term of FR allows the follower to move toward $r(x)$ at every iteration (finally it hits $r(x^*)$, see Figure 1), even in non-zero-sum games. As we argued in our previous responses, the correction term of FR can be thought of as a replacement of timescale separation. Though the exact dynamics of FR is different from $x_{t+1} = x_t - \gamma_1 h_1(x_t, r(x_t))$ (as we don't require y to be on the ridge $r(x)$), but it doesn't affect the fact that our FR converges to the right solution. And that's exactly our main contribution.
> > > > >
> > > > > To be clear, our FR dynamics is given by:
> > > > > $x_{t+1} = x_t - \gamma_1 h_1(x_t, y_t)$
> > > > > $y_{t+1} = y_t - \gamma_2 \nabla_y g(x_t, y_t) + \gamma_1 (\nabla_{yy} g)^{-1}\nabla_{yx}g  h_1(x_t, y_t)$
> > > > >
> > > > > To sum up, we are still not convinced that there is any need to use two timescale update in FR for our results to hold. We kindly ask the reviewer for providing other evidences to support his/her arguments. Otherwise, our theorem 3 should be taken as correct.

---

> > > > > > ### Comment · AnonReviewer1 · 2019-11-15
> > > > > > **Some Concerns Remain; Will leave it up to the AC**
> > > > > >
> > > > > > Thanks for the response.
> > > > > >
> > > > > > I want to reiterate that I think this paper studies an interesting problem, looks at an interesting set of dynamics, and has some impressive empirical results.
> > > > > >
> > > > > > However, in my opinion, either there is some problems with the results that are given for general-sum games or the assumptions and the result are not stated rigorously enough. It seems to me you are drawing conclusions about a system using y_t iterates which are not the same as r(x) and you have not shown that r(x) is attracting under the y_t dynamics surely. I can also try to say this another way. In the proof of theorem 3, you effectively plug in r(x) in the first line and after applying the similarity transform, in the upper left block, it appears you take the total derivative, as if y=r(x), to be able to get back the condition in prop 8, but you have not shown that y=r(x) under the FR dynamics. I think that under the right assumptions you could get the result you have, but it has been left up to some question since theorem 3 is stated in a vague manner.
> > > > > >
> > > > > > Again, thanks for being willing to have a cordial discussion. I hope that the prolonged back and forth has helped make your paper stronger in your view. For now I have left my score as is, and am willing to change during the reviewer discussion period if there is further evidence that the conclusion in the paper is fine.

---

> > > > > > > ### Author Response · Authors · 2019-11-15
> > > > > > > **The main concern has been addressed.**
> > > > > > >
> > > > > > > Thank you for your response and valuable input over the discussion period. We believe the discussion has benefited both of us. However, we respectfully disagree with your assessment of our general-sum results. In particular, your main concern of our result is already answered above.
> > > > > > >
> > > > > > > - Regarding why Theorem 3 uses $y=r(x)$, but we can reason about dynamics (8)
> > > > > > >
> > > > > > > We believe we have answered this question already in the paragraph "Connecting the result of Theorem 3 to our local convergence claim". Essentially, we can do this is because we can use the first-order Taylor series at the fixed point to characterize the behavior of the algorithm in the neighborhood of the fixed point.  Thus, we only need to compute the Jacobian at fixed points in Theorem 3. We would like to know exactly which part of this argument you find not convincing.
> > > > > > >
> > > > > > > - Regarding the statement of Theorem 3:
> > > > > > >
> > > > > > > We believe our statement of Theorem 3 is clear. The stability of fixed points is defined in Definition 2, and our requirement of the learning rate is given explicitly in the proof. Should you find the statement of Theorem 3 vague, we would like to ask you to point out exactly what requires further clarification.
> > > > > > >
> > > > > > > We would like to ask you to point out what exactly is wrong *in our reasoning*. We also think that stating your doubts clearly and exactly would greatly benefit the following reviewer discussion period.

---

> > > > > ### Author Response · Authors · 2019-11-15
> > > > > **More clarifications. FR is sort of a replacement of timescale separation.**
> > > > >
> > > > > As we notice in your last response, it seems that you are still reasoning about the issues of GDA + best-response gradient dynamics. Actually, that's exactly our motivation to propose FR. FR and timescale separation are two solutions to overcome the undesirable convergence of GDA + best-response gradient dynamics (in main paragraph, it was stated in the setting of zero-sum games). That also explains why we don't need two timescale separation. We stress again that FR is sort of a better replacement of timescale separation.
> > > > >
> > > > > Let us know if you have any other questions/comments. Hope we can reach consensus soon. And if you agree that our explanation makes sense and therefore theorem 3 holds. Please update your rating as you acknowledged that our method is novel and our contributions are significant.

---

> > > ### Author Response · Authors · 2019-11-14
> > > **Further Clarifications and Comparisons with [1]. The reason why we don't need timescale separation to work.**
> > >
> > > As the reviewer is concerned with our FR algorithm in general-sum games, we'd like to further clarify the relationship between our FR, timescale separation and best-response gradient dynamics (i.e. the dynamics of [1]). Hopefully, we can reach consensus before rebuttal deadline. Again, we appreciate all you detailed comments and suggestions.
> > >
> > > First, we present a view that FR in general-sum games is applying an additional preconditioner to the best-response gradient dynamics so as to fix the second-order condition of local Stackelberg equilibrium.
> > >
> > > From the sufficient/necessary conditions of local Stackelberg equilibrium, it can be seen that using best-response gradient to update is a way to match the first-order conditions. Therefore, it can be argued that best-response gradient dynamics is in fact the counterpart of GDA in general-sum games. Comparing FR in zero-sum games and non-zero-sum games ((5) and (8)), one can see that FR in non-zero-sum games essentially applies the same preconditioning matrix to best-response gradient dynamics. When applied to GDA in zero-sum games, the preconditioner fixes the second-order condition for local minimax and eliminates the need for timescale separation (Theorem 1). What we show in Theorem 3 is that, when applied to best-response gradient dynamics, it fixes the second-order condition for local Stackelberg equilibrium. From this point of view, it is reasonable why our algorithm does not require a timescale separation.
> > >
> > > In short, our FR algorithm adds a correction term to the follower's update (or equivalently applys an asymmetric preconditioner for the whole dynamics), which can be thought as a replacement of timescale separation rather than the best-response gradient term (which is to fix the first-order condition). We believe that our FR is better than timescale separation in the sense that 1. timescale separation can lead to slow convergence due to the use of small learning rate. By contrast, we are allowed to use large learning rates while still converge to the right solution. 2. FR in general-sum games converges exactly to local Stackelberg equilibria, whereas best-response graident dynamics with timescale separation [1] is not yet shown to have this desirable property (see Remark 3 in [1]).
> > >
> > > Next, we would like to discuss a potential reason why the best-response dynamics in [1] works for zero-sum-games (in the sense that stable fixed points are exactly local Stackelberg equilibria) without timescale separation, but not necessarily in general-sum-games. Our understanding is that, in zero-sum games, the best-response term $\nabla_{xy}^2 g (\nabla_{yy} g)^{-1} \nabla_y f$ becomes $\nabla_{xy}^2 f(\nabla_{yy} f)^{-1} \nabla_y f$. This provides the best-response gradient dynamics with a nice alternative explanation, namely predicting the gradient of $\nabla_{x} f(x,r(x))$ by estimating $y-r(x)$ to be $(\nabla_{yy}f)^{-1}\nabla_yf$ (a Newton step). It can be seen that this alternative explanation is no longer valid for general-sum games, since $y-r(x)\approx (\nabla_{yy}g)^{-1}\nabla_y g$, whereas the best-response term uses $ (\nabla_{yy} g)^{-1} \nabla_y f$. This coincident is also reflected in the Jacobian calculation (see (4) in [1]): the upper-right block of the Jacobian exactly cancels to zero.
> > >
> > > To sum up, we believe the key novelty of our algorithm is orthogonal to best-response gradient dynamics. Instead, we view it as a way to match the second-order conditions for local minimax/Stackelberg equilibria by combining it with GDA or best-response gradient dynamics. In the zero-sum setting, using best-response gradient dynamics indeed works (as shown in [1]). However, if our conjectured explanation is true, this could be somewhat coincidental.

---

> ### Author Response · Authors · 2019-11-15
> **Final BEST attempt. We updated the statement around Theorem 3.**
>
> On the one hand, we thank you for all detailed feedback and comments. We really appreciate your consistent dialogue with us. Your comments definitely helped make our paper stronger. On the other hand, we are a bit frustrated and upset. We believe we have addressed all your concerns so far, especially your concern about the validity of Theorem 3. It's a bit unfair to us since we cannot further participate in the reviewer discussion period. However, we won't give up and would like to give it our final best try. Hope we can reach consensus before rebuttal deadline.
>
> In your last response, you raised concern with that fact that we used $y = r(x)$ in our proof of Theorem 3. We would clarify that in the proof of Theorem 3, we are indeed just calculating the Jacobian at a fixed point (we can therefore safely use $y = r(x)$, see the statement before the calculation of Jacobian in page 16). Theorem 3 itself only says about stability of fixed points, which is a proxy for convergence. For local convergence, we have to combine it with Proposition 4 (see the newly added Remark 2). We apologize that we didn't refer to Proposition 4 in previous version, we've updated the paper to make this connection explicit.
>
> We stress again that the key is that we are allowed to do Taylor expansion around the fixed point (see Proposition 4). Therefore, we only need to calculate the Jacobian matrix for fixed points in Theorem 3. The combination of Theorem 3 and Proposition 4 implies local convergence (see Remark 2).
>
> We believe our paper deserves a score of 6 or even higher given our theoretical and empirical contributions. We've resolved all your concerns so far, and we hope you can reconsider your rating.

---

### Official Review · AnonReviewer3 · 2019-10-24
**Official Blind Review #3**

**Rating:** 6

**Review:**

In this paper, the authors introduce a new optimization algorithm for minimax problems, or finding equilibria in sequential
two-player zero-sum games. Such problems are common in machine learning, including generative adversarial networks or primal-dual reinforcement learning. The commonly used gradient descent-ascent algorithm, corresponding to taking a gradient step for both players (or for both variables being minimized and maximized over), does not converge, in general, to local minimax points. Moreover, it can converges to fixed points which are not local minimax. To address these issues, the authors introduce the "follow the ridge" algorithm for minimax optimization problems. Given a minimax problem min_x max_y f(x, y), this algorithm consists in adding a correction term to the gradient corresponding to the y variable (corresponding to the max). This term is derived from the observation that minimax optimization should follow ridges (i.e. local maximum w.r.t. to y) of the function. Ridges can be defined as the implicit functions such that the gradient w.r.t. y is equal to zero, allowing to design an update that would stay "close" to the ridge. The correction term corresponding to the update thus involve the inverse of the Hessian w.r.t. y. The authors prove that all the fixed points of this algorithm are minimax, and that all local minimax are fixed points of the algorithm. The proof use first and second order conditions for local minimax points, which were recently derived in a paper by Jin et al. The proposed algorithm can also be used with momentum and preconditioning, and be generalized to Stackelberg games. Finally, the authors evaluate the follow the ridge algorithm on toy low dimensional GAN problems, as well as experiments on the MNIST dataset, showing better convergence that other methods used for minimax optimization problem.

The problem studied in this paper is an important one, as it arises in multiple area of machine learning such as adversarial
training or reinforcement learning. It has also received significant attention from the community in the recent years. This paper propose a simple solution, which is well motivated, to the problem as well as a proof of convergence. A limitation of the proposed method is that it uses the Hessian of the problem, probably making it hard to apply on large  scale problems that are common in deep learning. I believe that it would make the paper stronger to discuss potential ways to mitigate this issue (e.g. inspired by L-BFGS), and their impact on theoretical guarantees. (Note that the authors briefly mention using the conjugate gradient algorithm in the experimental section).

Overall, the paper is well written, and easy to follow (even for non-expert like me). I believe that it does a good job at introducing the problem and existing work on which it builds, and to motivate the proposed solution. I have not checked the proofs carefully, but they seem sensible. A small weakness of the paper is the experimental section: for example, I am not sure the MNIST experiments bring much to the paper, and would have preferred more convincing experiments. However, this is mostly a theoretical paper, and I do not think this is a big concern.

To summarize, I think the paper study an important problem, proposes a sound solution and is clearly written. For these reasons, I believe that the paper should be accepted to the ICLR conference. However, as I am not an expert on this area, my recommendation is a low confidence one.


Minor comment: I believe that at the beginning of second paragraph of section 4, "Suppose that y_t is a local minimum of f(x_t, .)" should be "maximum".

**Experience Assessment:**

I do not know much about this area.

**Review Assessment: Checking Correctness Of Derivations And Theory:**

I assessed the sensibility of the derivations and theory.

**Review Assessment: Checking Correctness Of Experiments:**

I assessed the sensibility of the experiments.

**Review Assessment: Thoroughness In Paper Reading:**

I read the paper thoroughly.

---

> ### Author Response · Authors · 2019-11-06
> **Response to Reviewer #3**
>
> Thank you very much for your kind words about our work. It's really encouraging that you think the problem we study is important.
>
> Regarding the use of Hessian (and its inverse) in our method, we agree that it seems hard to generalize to large-scale machine learning tasks for exact Hessian computation. However, we note that conjugate gradient method (or Hessian-free method) only involves Hessian-vector product which has roughly the same computation cost as one backpropagation. Particularly, conjugate gradient has been successfully applied to a wide range of tasks such as reinforcement learning[2], image classification[3] and meta learning[4]. In the paper, we used conjugate gradient for GAN training and we've added more details in the Appendix. Lastly, we would like to note that Hessian is not necessary for standard supervised learning tasks since first-order methods like gradient descent converges to the right solution (local minima)[1]. However, it might not be the case for sequential games, we believe that the use of Hessian is necessary for problem we study, otherwise we might find a wrong solution. In terms of the approximation error of conjugate gradient and how it affects our convergence guarantees, we leave it for future work.
>
> [1] Lee et al., "Gradient descent only converges to minimizers", 2017
> [2] Schulman et al., "Trust Region Policy Optimization", 2015
> [3] Martens, "Deep learning via Hessian-free optimization", 2010
> [4] Rajeswaran et al., "Meta-Learning with Implicit Gradients", 2019

---

### Author Response · Authors · 2019-10-17
**Small visualization problem in Fig. 4**

We noticed a small visualization problem in Fig. 4. The KDE plots (first row) were generated by the seaborn package. The function kdeplot (https://seaborn.pydata.org/generated/seaborn.kdeplot.html) chooses a Gaussian kernel with improper bandwidth by default, so the modes in our figures look wider than they actually are. We emphasize that this does not affect our claim that FR learns the true distribution and our comparison of FR with other algorithms.

---

### Author Response · Authors · 2019-11-06
**First Revision**

We thank each of the reviewers for their time and comments.

We have uploaded our first revised version of our paper which addresses main concerns of reviewer #1. Particularly, we added details of our damping scheme in CG (Appendix D.4) and detailed derivation of proposition 7 and 8. Besides, we also removed section 4.1 due to the mistake reviewer #1 spotted.

---

### Decision · Program_Chairs · 2019-12-19

**Decision:**

Accept (Poster)

**Comment:**

The submission proposes a novel solution for minimax optimization which has strong theoretical and empirical results as well as broad relevance for the community. The approach, Follow-the-Ridge, has theoretical guarantees and is compatible with preconditioning and momentum optimization strategies.

The paper is well-written and the authors engaged in a lengthy discussion with the reviewers, leading to a clearer understanding of the paper for all. The reviews all recommend acceptance.